# FOXP1 phosphorylation antagonizes its O-GlcNAcylation in regulating ATR activation in response to replication stress

Xuefei Zhu [1,2,8] ✉, Congwen Gao [2,3,8], Bin Peng[1,2,8], Jingwei Xue[2], Donghui Xia[2,4,5], Liu Yang[2], Jiexiang Zhang[2,6], Xinrui Gao[2], Yilin Hu[7], Shixian Lin [7], Peng Gong [1] ✉ & Xingzhi Xu [1,2] ✉

## Abstract

ATR signaling is essential in sensing and responding to the replication stress; as such, any defects can impair cellular function and survival. ATR itself is activated via tightly regulated mechanisms. Here, we identify FOXP1, a forkhead-box-containing transcription factor, as a regulator coordinating ATR activation. We show that, unlike its role as a transcription factor, FOXP1 functions as a scaffold and directly binds to RPA–ssDNA and ATR–ATRIP complexes, facilitating the recruitment and activation of ATR. This process is regulated by FOXP1 O-GlcNAcylation, which represses its interaction with ATR, while CHK1-mediated phosphorylation of FOXP1 inhibits its O-GlcNAcylation upon replication stress. Supporting the physiological relevance of this loop, we find pathogenic FOXP1 mutants identified in various tumor tissues with compromised ATR activation and stalled replication fork stability. We thus conclude that FOXP1 may serve as a potential chemotherapeutic target in related tumors.

**Keywords** FOXP1; ATR; CHK1; Phosphorylation; O-GlcNAcylation
**Subject Categories** Chromatin, Transcription & Genomics; DNA Replication, Recombination & Repair

## Introduction

Replicating DNA is challenged by recurring endogenous or exogenous threats, such as ultraviolet, toxins to DNA polymerases or topoisomerases, base-alkylating agents, DNA crosslinkers, and dNTP synthesis inhibitors, which can lead to the stresses with slowing or stalling of replication forks (Lecona and Fernandez-Capetillo, 2018; Zeman and Cimprich, 2014). These replication stresses should be fully resolved in a timely manner to avoid the breakage or collapse of stalled replication forks, and ensure the faithful duplication of genomic information. Replication stress responses are regulated by sophisticated signaling networks, perhaps the most notable being the ATR (ATM- and Rad3-related)–CHK1 pathway (da Costa et al, 2023; Zeman and Cimprich, 2014). As a pivotal kinase triggered by replication stress, ATR catalyzes the phosphorylation of a series of substrates to initiate effective molecular events for maintaining genome stability. Such events include stabilizing and restarting stalled replication forks, repressing origin firing, balancing dNTP synthesis, and activating the cell cycle checkpoint. The latter is executed by CHK1 and serves to buy time for stress resolution before cells enter mitosis (Saldivar et al, 2017). Mechanistically, ATR and its partner ATRIP, is recruited to RPA-coated single-stranded DNA (ssDNA), where it is activated and binds with two allosteric activators, TopBP1 and ETAA1 (Bass et al, 2016; Delacroix et al, 2007; Haahr et al, 2016; Lee et al, 2007; Tannous et al, 2021; Zou and Elledge, 2003).

ATR activation and the subsequent downstream signaling that ensues are tightly regulated via posttranslational modifications (PTMs), such as phosphorylation, ubiquitination, SUMOylation, and acetylation. Phosphorylation is perhaps the best-characterized modification implicated in ATR-CHK1 signaling, as ATR and CHK1 exhibit full kinase activity only after being phosphorylated; only then can ATR and CHK1 mediate the phosphorylation of other substrates to trigger replication fork protection and the cell cycle checkpoint (da Costa et al, 2023). O-GlcNAcylation is another important PTM; here, O-linked N-acetylglucosamine (O-GlcNAc) is reversibly attached and removed from serine or threonine of substrates by O-GlcNAc transferase (OGT) and O-GlcNAcase (OGA), respectively (Bond and Hanover, 2015). Perturbations to O-GlcNAcylation levels can greatly affect normal cellular physiological functions and even survival (Chatham et al, 2021). O-GlcNAcylation and phosphorylation share the same acceptor residues (serine and threonine), and thus may compete for the same

[1]Carson International Cancer Center & Department of General Surgery & Institute of Precision Diagnosis and Treatment of Gastrointestinal Tumors, Shenzhen University General Hospital, Shenzhen University Medical School, 518060 Shenzhen, Guangdong, China. [2]Guangdong Key Laboratory for Genome Stability & Disease Prevention and Marshall Laboratory of Biomedical Engineering, Shenzhen University Medical School, 518060 Shenzhen, Guangdong, China. [3]College of Life Sciences, Institute of Life Sciences and Green Development, Hebei University, 071002 Baoding, China. [4]Shenzhen University General Hospital-Dehua Hospital Joint Research Center on Precision Medicine (sgh-dhhCPM), Dehua Hospital, Dehua, 362500 Quanzhou, China. [5]State Key Laboratory of Agro-biotechnology and MOA Key Laboratory of Soil Microbiology, College of Biological Sciences, China Agricultural University, 100193 Beijing, China. [6]Shenzhen University-Friedrich Schiller Universität Jena Joint PhD Program in Biomedical Sciences, Shenzhen University School of Medicine, 518060 Shenzhen, Guangdong, China. [7]Life Sciences Institute, Zhejiang University, 310058 Hangzhou, China. [8]These authors contributed equally: Xuefei Zhu, Congwen Gao, Bin Peng. ✉E-mail: zhuxuefei@szu.edu.cn; doctorgongpeng@szu.edu.cn; Xingzhi.Xu@szu.edu.cn

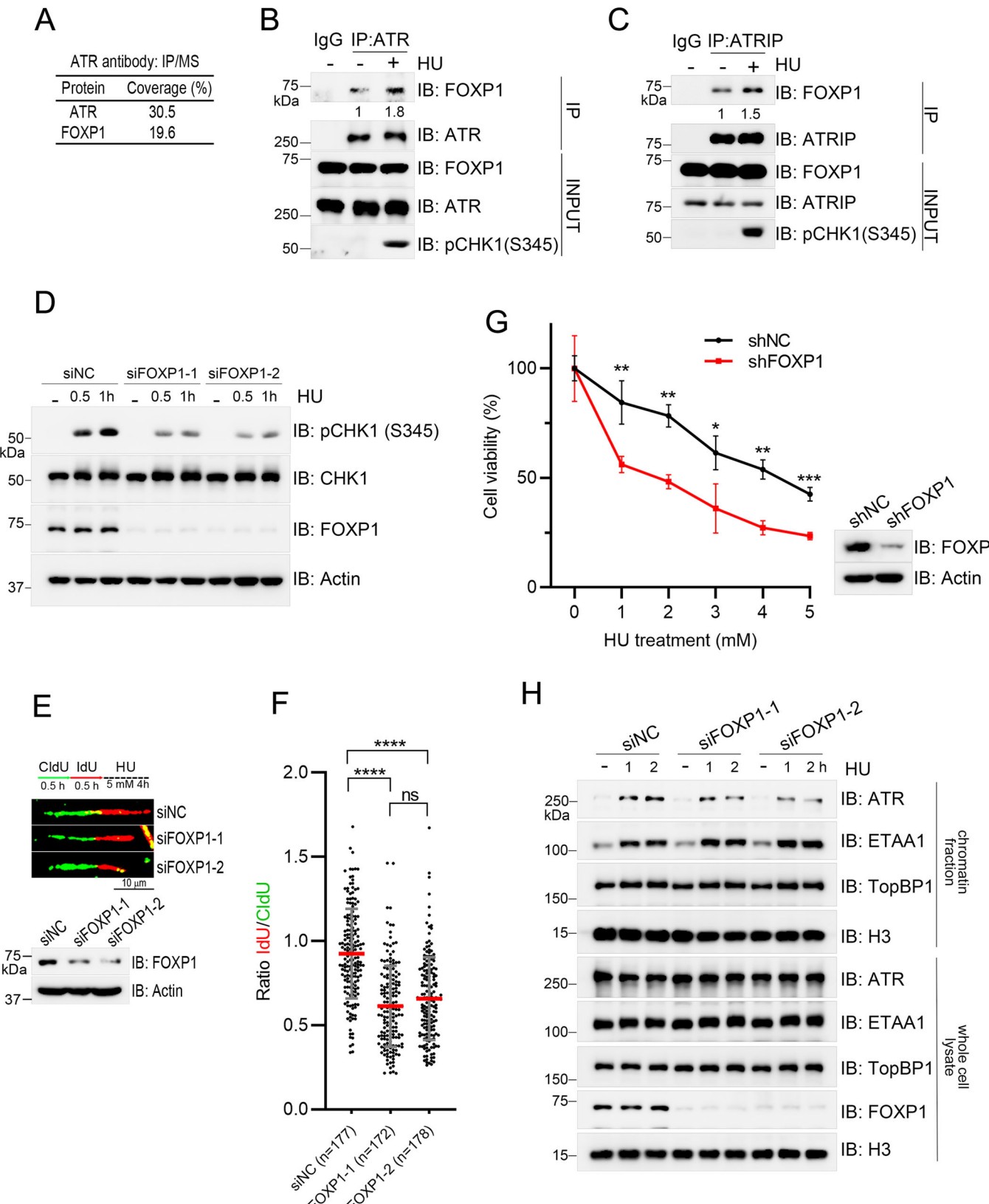

**Figure 1. FOXP1 promotes ATR activation under replication stress.**

(A) HEK293T cells were subjected to immunoprecipitation using ATR-specific antibody, the immunoprecipitates were analyzed via mass spectrum and FOXP1 was identified. (B, C) HEK293T cells were treated with 2 mM HU for 1 h or left untreated before immunoprecipitation using ATR (B) or ATRIP (C) specific antibodies. The immunoprecipitates were analyzed via immunoblotting using the indicated antibodies. (D) HEK293T cells transfected with a negative control siRNA or different siRNAs targeting FOXP1 were treated with HU for the indicated time before the whole cell lysates were harvested for immunoblotting with the indicated antibodies. (E) Upper panel: schematic of the DNA fiber assay examining stalled replication fork stability. Middle: representative images of CldU and IdU replication tracks. Lower panel: FOXP1 levels in different H1975 cells. (F) Statistical analysis of the IdU/CldU ratio of DNA fibers; the IdU/CldU ratio mean (red line) ±SD is shown. $n$, DNA fiber number, ****$P < 0.0001$; ns no significance; $P$ values were calculated by one-way ANOVA, followed by Kruskal–Wallis test. $P$ value: siNC vs siFOXP1-1, 1.65e-024; siNC vs siFOXP1-2, 2.48e-018; siFOXP1-1 vs siFOXP1-2, 0.3593. (G) Control and FOXP1 knockdown H1975 cells were incubated with the indicated doses of HU for 6 h and then cultured for 14 d. The colonies were then stained with crystal violet and the percentage of viable cells were calculated. The mean percentage of viable cells (biological replicates, $n = 3$) ± SD is shown. *$P < 0.05$; **$P < 0.01$; ***$P < 0.001$, $P$ values were calculated by unpaired two-tailed $t$ test. $P$ value, shNC vs shFOXP1: 1 mM, 0.0098; 2 mM, 0.0011; 3 mM, 0.0320; 4 mM, 0.0011; 5 mM, 0.0008. (H) HEK293T cells transfected with a negative control siRNA or different siRNAs targeting FOXP1 were treated with 2 mM HU for the indicated time, and then subjected to chromatin fractionation. The protein levels in the whole cell lysate and chromatin fractions were examined via immunoblotting using the indicated antibodies. Source data are available online for this figure.

modified site on one substrate, or these two modifications may simultaneously or reciprocally occur on different sites of one substrate (Chatham et al, 2021; Takayama et al, 2008; Zeidan and Hart, 2010). O-GlcNAcylation level in cells seems to correlate with ATR-CHK1 signaling (Na et al, 2020). Indeed, OGT is phosphorylated by CHK1 for stabilization (Li et al, 2017), and elevated global O-GlcNAcylation increases the phosphor-S/TQ signal, which is specifically catalyzed by ATR or ATM kinases (Na et al, 2020). As OGT is the only O-GlcNAc transferase identified to date (Chatham et al, 2021), considering its substrate variety, the precise correlation between protein O-GlcNAcylation and ATR signaling is poorly understood.

In this study, we identified FOXP1 as an ATR-interacting protein and a regulator in promoting ATR activation, under the regulation of FOXP1 O-GlcNAcylation and phosphorylation. The interplay between CHK1-mediated FOXP1 phosphorylation and OGT-mediated FOXP1 O-GlcNAcylation forms a feed-forward loop in facilitating the ATR-CHK1 signaling activation. The necessity of functional FOXP1 is clear, given that FOXP1 knockout leads to embryonic lethality in mice, while FOXP1 haploinsufficiency leads to developmental retardation and tumorigenesis (Gao et al, 2023; Rappold et al, 1993; Wang et al, 2004), these symptoms caused by FOXP1 deficiency are similar to that caused by ATR signaling deficiency (Zeman and Cimprich, 2014). Our findings reveal a mechanism by which the transcription factor FOXP1 can also assume a structural role to help facilitate ATR signaling in response to replication stress, and that also potentially underlies the pathogenesis of FOXP1 deficiency.

## Results

### FOXP1 promotes ATR activation in response to replication stress

ATR signaling is under regulation of complex mechanisms, and the activation of ATR is not fully understood. We thus set out to identify any new regulators of ATR activation in response to replication stress. To do so, we performed immunoprecipitation using an ATR-specific antibody in HEK293T cells, followed by mass spectrum analysis. One of the ATR immunoprecipitates we detected was FOXP1 (Figs. 1A and EV1A), a DNA binding protein whose characterized functions are related to transcription

regulation (Gao et al, 2023), and here, FOXP1 might function beyond a transcription factor. FOXP1 haploinsufficiency is identified in development disorders and tumor tissues, these symptoms are similar to that caused by ATR signaling, promoting us to explore the relationship between FOXP1 and ATR signaling. We verified the interaction by endogenous immunoprecipitation again using an ATR-specific antibody in HEK293T cells (Fig. 1B). Meanwhile, we also queried whether this interaction is responsive to replication stress, and thus tested the FOXP1 and ATR interaction under treatment with hydroxyurea (HU). Here we saw that the interaction between FOXP1 and ATR was increased upon HU-induced replication stress (Fig. 1B). As ATR and ATR-interacting protein (ATRIP) form a complex, we also tested the interaction between FOXP1 and ATRIP via endogenous immunoprecipitation using an ATRIP-specific antibody in HEK293T cells, with or without HU exposure. We also detected an increased FOXP1 and ATRIP interaction upon replication stress (Fig. 1C). We next wanted to know whether FOXP1 directly interacts with the ATR–ATRIP complex. Indeed, we detected that His-tagged FOXP1 directly interacted with GST-tagged ATRIP in vitro (Fig. EV1B); however, we failed to purify the ATR protein, so we can not rule out the possibility that FOXP1 also directly interacts with ATR as seen for TopBP1 that associates with both ATR and ATRIP (Kumagai et al, 2006).

Our results suggested that FOXP1 might be a novel regulator of ATR activation based on the increased interaction between FOXP1 and the ATR–ATRIP complex upon replication stress. To test our hypothesis, we monitored ATR activation (by examining ATR-mediated CHK1 phosphorylation at S345) in HEK293T cells and H1975 cells transfected with a negative control siRNA or an siRNA targeting FOXP1, and ATR-mediated CHK1 phosphorylation at S345 were compromised in FOXP1-depleted cells (Figs. 1D and EV1C). Seeing that FOXP1 contributed to the activation of ATR, we next queried whether FOXP1 depletion also affected stalled replication fork stability. To do so, we tested the stability of stalled replication forks via DNA fiber assay in H1975 cells transfected with a negative control siRNA or FOXP1 siRNA. We sequentially labeled the stages of DNA synthesis using the thymidine analogs CldU and IdU before exposing the cells to HU. We detected a decreased IdU/CldU ratio, indicating the instability of stalled replication forks in FOXP1-depleted cells compared with the control cells (Figs. 1E,F and EV1D). The FOXP1-depleted cells were also more sensitive to HU treatment, as shown by the reduction of cell viability compared with that of the control cells (Fig. 1G).

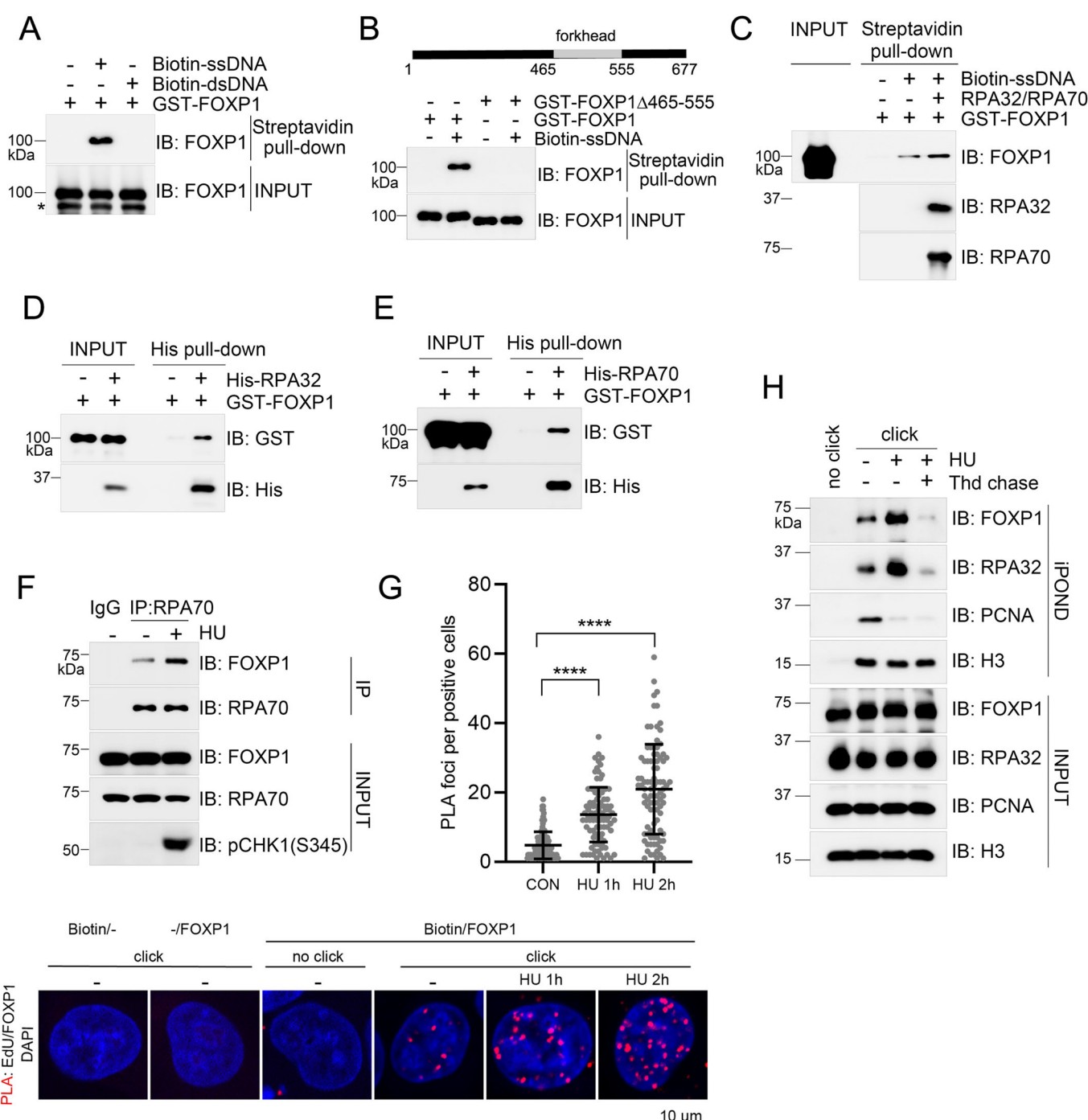

As replication stress-induced ATR activation occurs in S phase, we also examined the cell cycle distribution in cells with or without FOXP1 depletion, and found no difference in the S phase distribution in HEK293T or H1975 cells regardless of FOXP1 depletion (Fig. EV1E). Since FOXP1 promoted ATR-CHK1 activation under replication stress, we saw deficient cell cycle arrest at S phase in FOXP1-depleted cells comparing with that in control cells after incubation with low dose of camptothecin (CPT) in HEK293 cells (Fig. EV1F). Data from another study showed that FOXP1 promotes the expression of GINS1, a component of the

CDC45-MCM-GINS helicase that unwinds template DNA during replication, in diffuse large B-cell lymphoma (DLBCL) (Chen et al, 2023), but in HEK293T cells and H1975 cells, we saw no reduction on GINS1 levels in FOXP1-depleted cells (Fig. EV1G). We also examined DNA replication via DNA fiber assay, and also saw no effect as a result of FOXP1 depletion (Fig. EV1H). We further tested the chromatin loading of ATR upon replication stress in wild-type and FOXP1-depleted HEK293T cells by extracting the chromatin-bound proteins and detected compromised ATR levels in the chromatin fraction of FOXP1-depleted cells; however, the

**Figure 2.  FOXP1 loads onto stalled replication forks.**

(A) Biotin-labeled ssDNA or complemented dsDNA were conjugated on streptavidin magnetic beads and incubated with GST-tagged FOXP1 purified from *E. coli*. Streptavidin-bound FOXP1 was detected via immunoblotting with the indicated antibodies. *, degraded GST-FOXP1. (B) Biotin-labeled ssDNA conjugated on streptavidin magnetic beads were incubated with GST-tagged FOXP1 or a GST-tagged FOXP1 Δ465–555 mutant purified from *E. coli*. Streptavidin-bound FOXP1 was detected via immunoblotting with the indicated antibodies. (C) Biotin-labeled ssDNA conjugated on streptavidin magnetic beads, with or without preincubation with His-tagged RPA70/RPA32, were incubated with GST-tagged FOXP1. Streptavidin-bound proteins were detected via immunoblotting with the indicated antibodies. (D, E) His-tagged RPA32 (D) or His-tagged-RPA70 (E) were incubated with GST-tagged FOXP1 before His pull-down assay. Proteins bound onto Ni beads were detected via immunoblotting with the indicated antibodies. (F) HEK293T cells treated with 2 mM HU for 1 h or untreated were subjected to immunoprecipitation using a RPA70-specific antibody. The immunoprecipitates were analyzed via immunoblotting with the indicated antibodies. (G) Proximity ligation assay experiments using FOXP1 and biotin-specific antibodies in H1975 cells. Lower: representative images of PLA foci. upper: quantification of the number of PLA foci per foci-positive cells (cell number: CON, $n = 124$; HU-1h, $n = 95$; HU-2h, $n = 94$), mean ± SD is shown. ****$P < 0.0001$, $P$ values were calculated by one-way ANOVA, followed by Kruskal–Wallis test. $P$ value: CON vs HU 1 h, 7.20e-014; CON vs HU 2 h, 6.02e-027. (H) HEK293T cells were labeled with EdU followed by analysis by iPOND assay and immunoblotting with the indicated antibodies. Source data are available online for this figure.

chromatin loading of ETAA1 and TopBP1, the two allosteric activators of ATR, were not affected in FOXP1-depleted cells (Fig. 1H). These findings suggest that FOXP1 promotes ATR chromatin loading. Indeed, we detected that the interaction between FOXP1 and ATR mainly existed in the soluble cell fraction during unperturbed conditions, but more so in the chromatin fraction under conditions of replication stress (Fig. EV1I). Collectively, these findings indicate that FOXP1 may promote ATR activation by recruiting ATR onto chromatin.

## FOXP1 loads onto stalled replication forks

As FOXP1 is a DNA binding protein and we now know promotes ATR activation, we queried whether FOXP1 also loaded onto stalled replication forks. The stretch of ssDNA generated at stalled replication forks due to the uncoupling of DNA polymerases and CMG helicase, is a platform for the loading of a series of regulators and is protected by the RPA complex (Byun et al, 2005). We examined whether FOXP1 could directly bind to ssDNA using biotin-labeled ssDNA, which we enriched with streptavidin-conjugated magnetic beads and incubated with recombinant GST-tagged FOXP1 purified from *E. coli*. We detected that FOXP1 directly bound to ssDNA, however, the affinity was notably decreased when ssDNA was complemented to form double-stranded DNA (dsDNA), indicating that FOXP1 preferred to bind ssDNA in vitro (Fig. 2A). As a transcription factor, FOXP1 prefers to bind consensus motif GTAAACA on dsDNA (Gabut et al, 2011; Li et al, 2004). We wondered whether the binding of FOXP1 to ssDNA is also sequence-specific. We detected that FOXP1 has a preference for binding dsDNA containing the GTAAACA consensus (Fig. EV2A), while its affinity for ssDNA was similar, regardless of whether the sequence included the GTAAACA motif (Fig. EV2B). The FOXP1 forkhead domain (amino acids (AA) 465–555) confers it DNA binding capacity (Wang et al, 2003). We saw that this domain-mediated FOXP1's affinity for ssDNA, as deletion of AA 465–555 prevented the binding of FOXP1 with ssDNA (Fig. 2B).

Next, we incubated ssDNA with RPA70/32 before adding GST-tagged FOXP1 to the reaction mixture. Here we saw that FOXP1 exhibited a higher affinity for RPA-coated ssDNA than non-coated ssDNA (Fig. 2C). Indeed, we detected a direct interaction between the recombinant GST-tagged FOXP1 and His-tagged RPA70, or His-tagged RPA32 purified from *E. coli* (Fig. 2D,E). We also detected the interaction between FOXP1 and RPA70 via

endogenous immunoprecipitation in HEK293T cells using an RPA70-specific antibody; this interaction increased under conditions of HU-induced replication stress (Fig. 2F).

Finally, we wanted to narrow down the precise region by which FOXP1 interacts with RPA. To do so, we generated a series of FLAG-tagged FOXP1 deletion mutants and after immunoprecipitation in HEK293T cells expressing the full-length or those deletion mutants of FOXP1, we saw that AA 431–555 was responsible for the interaction between FOXP1 and RPA70/32 in vivo (Fig. EV2C,D). Refining this further, results of an in vitro GST pull-down assay showed that AA 465–555 directly mediated the interaction between FOXP1 and RPA70 or RPA32. Deletion of AA 451–464 had no effect on the direct interaction between FOXP1 and RPA proteins, and deletion of AA 431–450 only compromised its direct interaction with RPA70 (Fig. EV2E,F). To confirm the loading of FOXP1 on stalled replication forks, we labeled nascent DNA in H1975 cells with EdU, which was later conjugated and labeled with biotin-azide. We then performed a proximity ligation assay using biotin-specific and FOXP1-specific antibodies. We detected a co-localization signal between FOXP1 with biotin under unperturbed conditions, which was elevated upon HU-induced replication stress (Fig. 2G), indicating that FOXP1 accumulates at stalled replication forks. Next, we performed an isolation of proteins on nascent DNA (iPOND) to validate the loading of FOXP1 on stalled replication forks, and we saw low FOXP1 levels on replication forks, which was notably increased under conditions of replication stress induced by HU (Fig. 2H). As a direct RPA-binding protein, FOXP1 recruitment to stalled replication forks may be RPA-dependent, and we saw decreased loading of FOXP1 onto stalled replication forks in RPA-deficient cells (Fig. EV2G). Together, these data indicate that FOXP1 loads onto stalled replication forks by directly binding to RPA-coated ssDNA.

## OGT-mediated FOXP1 O-GlcNAcylation represses its interaction with ATR

Although the PTMs of FOXP1 remain largely unknown, we wondered whether the scaffold function of FOXP1 that serves to recruit ATR and promote its activation, is regulated by PTM(s). A previously generated mass spectrum dataset indicated that OGT potentially interacted with FOXP1 (Estruch et al, 2018), so we performed a series of experiments to validate their interaction and the O-GlcNAcylation of FOXP1. We first confirmed the interaction between FOXP1 and OGT via immunoprecipitation, detecting

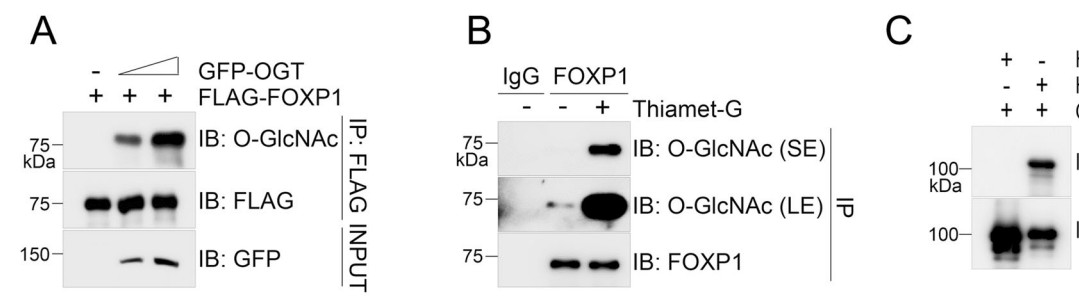

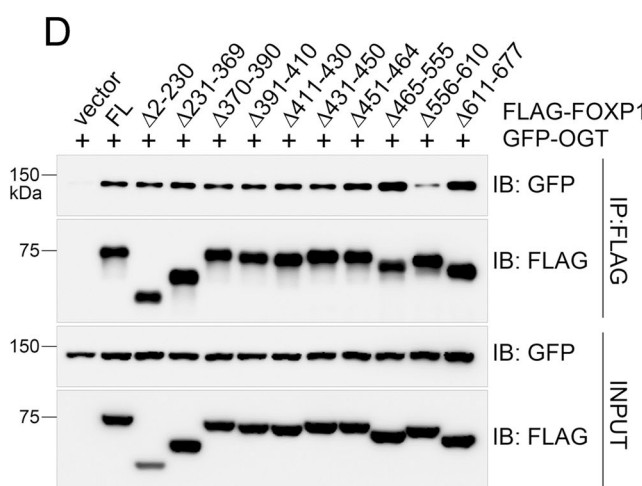

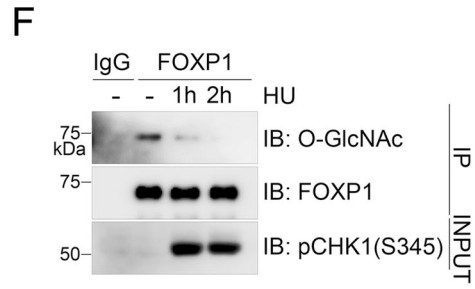

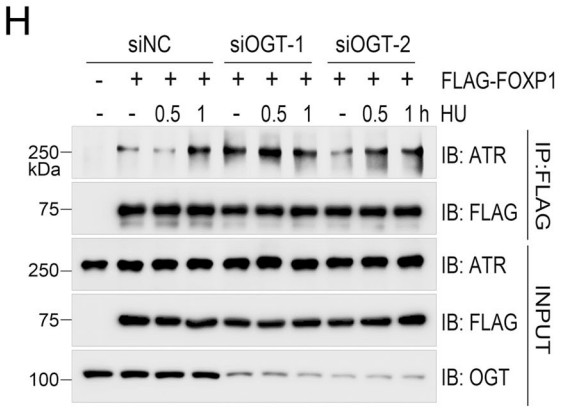

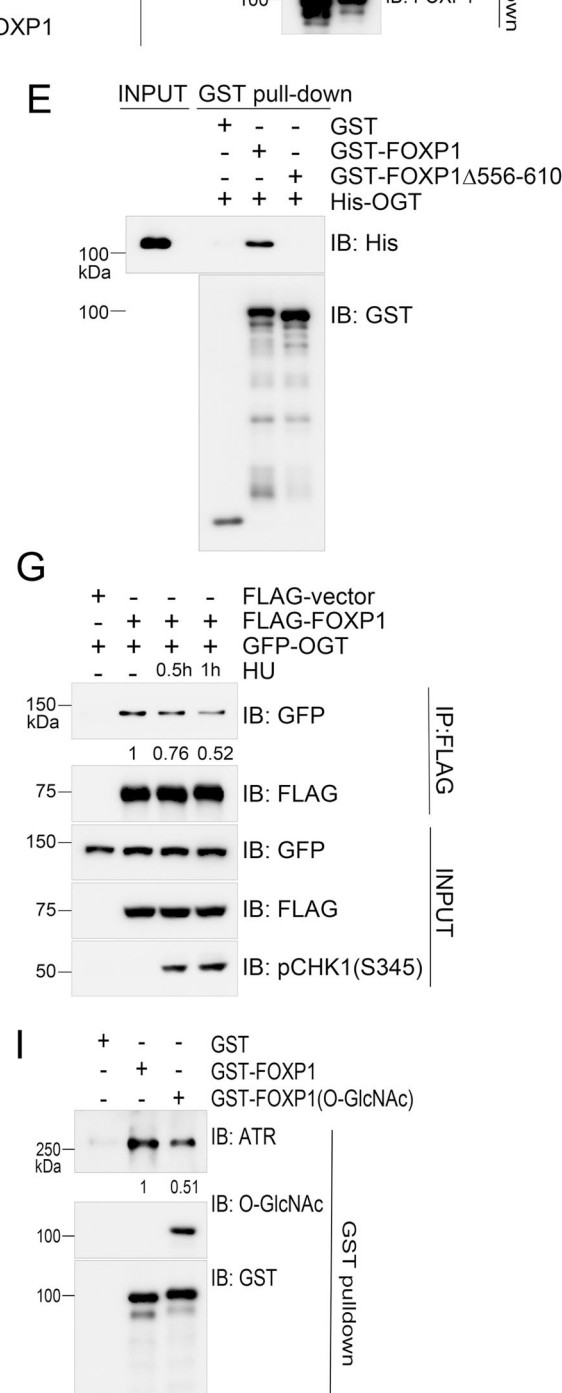

**Figure 3.   OGT-mediated O-GlcNAcylation of FOXP1 represses its interaction with ATR.**

(A) HEK293T cells transfected with FLAG-tagged FOXP1 and increasing concentrations of GFP-tagged OGT, were subjected to immunoprecipitation using a FLAG-specific antibody. The immunoprecipitates were examined via immunoblotting using the indicated antibodies. (B) HEK293T cells preincubated with Thiamet-G (5 μM, 24 h) or not were subjected to immunoprecipitation using a FOXP1-specific antibody. The immunoprecipitates were examined via immunoblotting using the indicated antibodies. (C) GST-tagged FOXP1 was purified from bacteria expressing GST-FOXP1 along with His-OGT or the corresponding empty vector. GST-FOXP1 and its O-GlcNAcylation were detected via immunoblotting with the indicated antibodies. (D) HEK293T cells transfected with GFP-tagged OGT and wild-type FLAG-tagged FOXP1 or its deletion mutants were subjected to immunoprecipitation using a FLAG-specific antibody. The immunoprecipitates were examined via immunoblotting using the indicated antibodies. (E) His-tagged OGT was incubated with wild-type GST-tagged FOXP1 or its mutant lacking AA 556–610, before performing a GST pull-down assay. Proteins bound onto glutathione beads were detected via immunoblotting using the indicated antibodies. (F) HEK293T cells treated with 2 mM HU for the indicated time were subjected to immunoprecipitation using a FOXP1-specific antibody. The immunoprecipitates were examined via immunoblotting using the indicated antibodies. (G) HEK293T cells transfected with FLAG-tagged FOXP1 and GFP-tagged OGT were treated with 2 mM HU for the indicated time and subjected to immunoprecipitation using a FLAG-specific antibody. The immunoprecipitates were examined via immunoblotting using the indicated antibodies. (H) Negative control or OGT knockdown HEK293T cells were transfected with FLAG-tagged FOXP1 and treated with 2 mM HU for the indicated time before immunoprecipitation using a FLAG-specific antibody. The immunoprecipitates were examined via immunoblotting using the indicated antibodies. (I) GST, GST-tagged FOXP1, or GST-tagged FOXP1 modified with O-GlcNAc were incubated with HEK293T cell lysate before GST pulldown. The glutathione bead-bound signals were detected by immunoblotting. * non-specific signal. Source data are available online for this figure.

OGT in the immunoprecipitates of FLAG-tagged FOXP1 (Fig. EV3A), and vice versa for SFB-tagged OGT (Fig. EV3B). We detected high levels of O-GlcNAc modification on FLAG-tagged FOXP1 enriched from HEK293T cells by immunoprecipitation (Fig. EV3C), and we saw that FOXP1 O-GlcNAcylation increased alongside the elevated expression level of GFP-tagged OGT (Fig. 3A). Elevated FOXP1 O-GlcNAcylation was also detected in the presence of Thiamet-G (Fig. 3B), an inhibitor of the only identified O-GlcNAcase, OGA (Chatham et al, 2021). Meanwhile, we co-expressed GST-tagged FOXP1 and His-tagged OGT in *E. coli*, and after collecting FOXP1 by GST pulldown, we could detect the O-GlcNAcylation of GST-tagged FOXP1 by immunoblotting (Fig. 3C). These data indicate that OGT directly catalyzes the O-GlcNAcylation of FOXP1.

We next mapped the interaction between FOXP1 and OGT and found that an uncharacterized domain on FOXP1 within AA 556–610, was responsible (Fig. 3D). After generating a series of deletion mutants Δ556–570, Δ571–590, and Δ591–610, we saw a compromised interaction with OGT to a similar extent as a Δ556–610 mutant (Fig. EV3D), indicating that the whole region (AA 556–610) is essential for an optimized interaction between FOXP1 and OGT. GST pull-down assay further confirmed the direct interaction between recombinant GST-tagged FOXP1 and His-tagged OGT purified from *E. coli*, and the interaction was mediated by AA 556–610 on FOXP1 (Fig. 3E).

We were interested to test the level of FOXP1 O-GlcNAcylation under conditions of replication stress. We thus treated HEK293T cells with HU and saw that FOXP1 O-GlcNAcylation was decreased under replication stress compared to steady-state conditions (Figs. 3F and EV3E), and increased replication stress-induced disassociation of FOXP1 from OGT (Fig. 3G). The phenomenon that replication stress increased the interaction between FOXP1 and ATR, yet decreased the interaction between FOXP1 and OGT, raised the possibility that FOXP1 O-GlcNAcylation helps to regulate the replication stress response.

We next tested the interaction between FOXP1 and ATR in HEK293T cells with different OGT levels, and found that OGT depletion with specific siRNAs promoted the interaction between FOXP1 and ATR compared with that in cells transfected with a negative control siRNA (Fig. 3H). We also saw that the interaction between FOXP1 and ATR decreased in the presence of the OGA inhibitor Thiamet-G (Fig. EV3F). To directly examine the effect of

FOXP1 O-GlcNAcylation on its interaction with ATR, we co-expressed GST-tagged FOXP1 with His-tagged OGT or an empty vector, and purified GST-tagged FOXP1 with or without O-GlcNAcylation to incubate with HEK293T cell lysates. The GST pull-down assay showed that O-GlcNAcylated GST-tagged FOXP1 exhibited a compromised interaction with ATR (Fig. 3I). We thus conclude that O-GlcNAcylation of FOXP1 represses its interaction with ATR, and that FOXP1 O-GlcNAcylation decreases while the interaction between FOXP1 and ATR increases during conditions of replication stress.

## CHK1-mediated FOXP1 phosphorylation at S396 antagonizes its O-GlcNAcylation

During our explorations into the ATR and FOXP1 association, we found that the interaction between FOXP1 and the ATR–ATRIP complex was dependent on ATR kinase activity under conditions of replication stress. Specifically, we saw that when incubating HEK293T cells exposed to HU with ATR inhibitors VE-822 or NU6027, the increased interaction between FOXP1 and the ATR–ATRIP complex was compromised (Fig. 4A). We thus wondered whether ATR phosphorylated FOXP1 to form a positive feedback loop to regulate ATR activation. However, we failed to detect any phosphor-S/TQ signal on FOXP1 both in unperturbed or replication stress states, so FOXP1 might not be an ATR substrate, and ATR kinase activity might affect FOXP1 function indirectly through ATR substrate. We next queried whether the interaction between FOXP1 and the ATR–ATRIP complex was regulated by the ATR substrate CHK1. Meanwhile, we found that the replication stress-induced interaction between FOXP1 and the ATR–ATRIP complex was also dependent on CHK1 kinase activity, as the increased association between FOXP1 and ATR–ATRIP complex was compromised in the presence of CHK1 inhibitors Rabusertib or UCN-01 (Fig. 4B). Moreover, replication stress-induced de-O-GlcNAcylation of FOXP1 was prevented upon simultaneous treatment with CHK1 inhibitor (Fig. 4C).

Due to the intimate crosstalk between phosphorylation and O-GlcNAcylation, we wondered whether FOXP1 was a potential CHK1 substrate. Indeed, we detected an interaction between FOXP1 and CHK1 via endogenous immunoprecipitation using a CHK1-specific antibody, and the interaction was slightly increased under conditions of replication stress (Fig. EV4A). We further

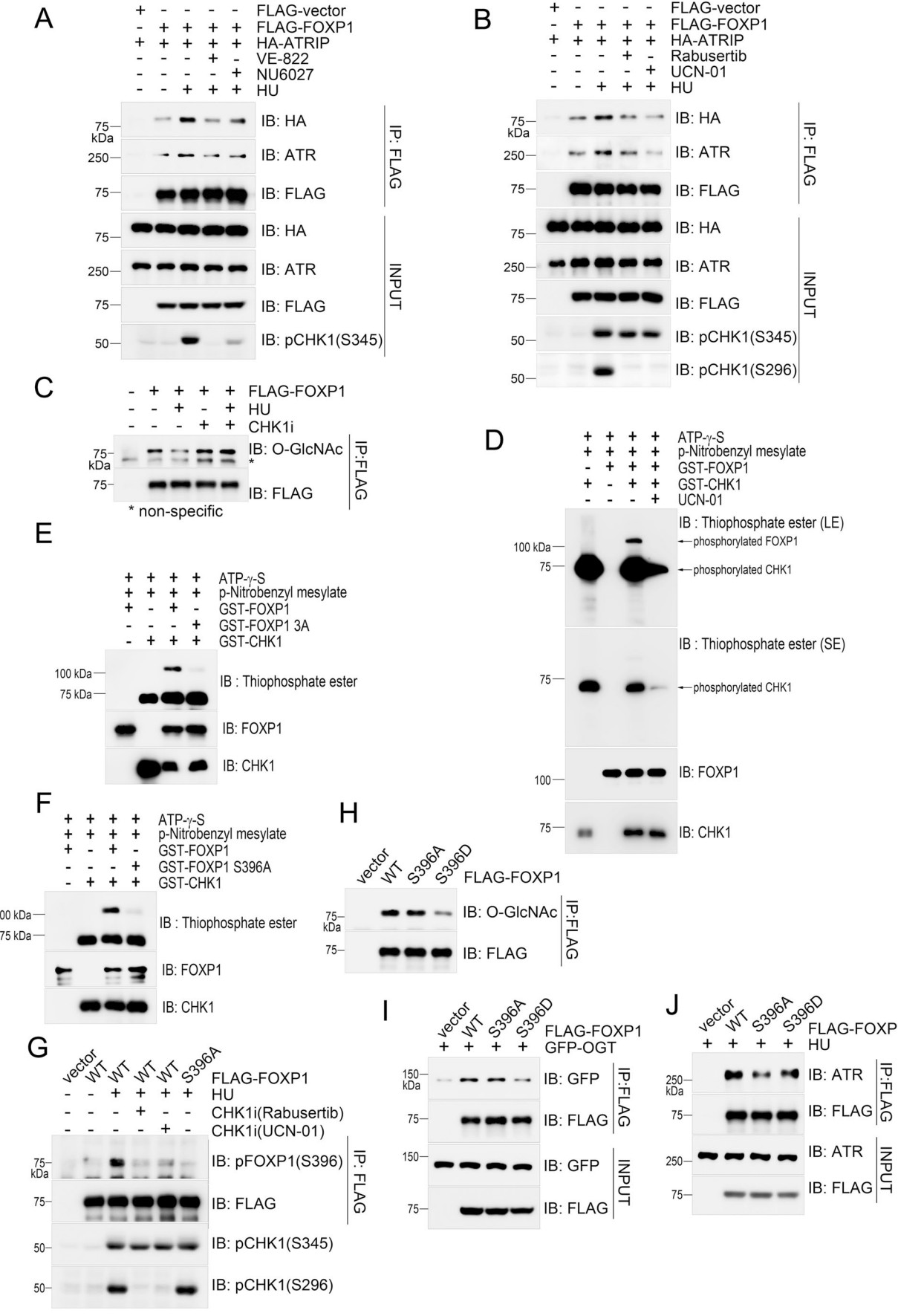

**Figure 4. CHK1-mediated FOXP1 phosphorylation at S396 antagonizes its O-GlcNAcylation.**

(A, B) HEK293T cells transfected with FLAG-FOXP1 and HA-ATRIP treated with 2 mM HU for 1 h in the presence of ATR inhibitors VE-822 (1 μM) and NU6027 (10 μM) (A) or CHK1 inhibitors Rabusertib (5 μM) and UCN-01 (50 nM) (B), were subjected to immunoprecipitation using a FLAG-specific antibody. The immunoprecipitates were analyzed via immunoblotting using the indicated antibodies. (C) HEK293T cells overexpressing FLAG-tagged FOXP1 were treated with 2 mM HU for 1 h in the presence of the CHK1 inhibitor Rabusertib or not, before the cells were subjected to immunoprecipitation using a FLAG-specific antibody. O-GlcNAcylation of FOXP1 was detected by immunoblotting. *, non-specific signal. (D) An in vitro kinase assay reaction mixture consisting of GST-tagged CHK1, GST-tagged FOXP1, and the phosphor group donor ATR-γ-S, with or without CHK1 inhibitor UCN-01, was incubated for 0.5 h, and further incubated with p-Nitrobenzyl mesylate for 2 h. The phosphorylation signals were detected via immunoblotting using a thiophosphate ester-specific antibody. (E) In vitro phosphorylation of FOXP1 and its T236A/S396A/S440A mutant by CHK1 was established as described in (D). (F) In vitro phosphorylation of FOXP1 and its S396A mutant by CHK1 was established as described in (D). (G) HEK293T cells transfected with FLAG-tagged FOXP1 treated with 2 mM HU for 1 h with or without CHK1 inhibitors, Rabusertib (5 μM) or UCN-01 (50 nM), were subjected to immunoprecipitation using a FLAG-specific antibody. The immunoprecipitates were examined via immunoblotting using a specific antibody to detect FOXP1 phosphorylation at S396. (H) HEK293T cells overexpressing wild-type FLAG-tagged FOXP1, its phosphorylation deficient mutant S396A, or its phosphorylation mimic mutant S396D were subjected to immunoprecipitation using a FLAG-specific antibody. O-GlcNAcylation of FOXP1 was detected via immunoblotting. (I) HEK293T cells overexpressing GFP-tagged OGT and wild-type FLAG-tagged FOXP1, S396A or S396D mutants were subjected to immunoprecipitation and immunoblotting using the indicated antibodies. (J) HEK293T cells transfected with wild-type FLAG-tagged FOXP1, S396A, or S396D mutant were treated with 2 mM HU for 1 h and subjected to immunoprecipitation and immunoblotting using the indicated antibodies. Source data are available online for this figure.

performed an in vitro phosphorylation assay to test CHK1-mediated phosphorylation of FOXP1. In this assay, we used ATP-γ-S as the phosphor group donor, and with p-Nitrobenzyl mesylate for alkylation of the thio-phosphorylation group, which was recognized by a thiophosphate ester-specific antibody via immunoblotting. Indeed, we detected that CHK1 did phosphorylate FOXP1, and as a confirmation, the signals for phosphorylated FOXP1 and auto-phosphorylated CHK1 were abolished in the presence of the CHK1 inhibitor UCN-01 (Fig. 4D).

CHK1 specifically recognizes serine or threonine in a conserved Φ-X-β-X-X-(S/T) (Φ: hydrophobic residue; β: basic residue) motif (Hutchins et al, 2000). FOXP1 contains three residues that can be potentially phosphorylated by CHK1: T236, S396, and S440 (Fig. EV4B). We mutated all these three residues to alanine, and then monitored FOXP1 phosphorylation by CHK1. We detected compromised phosphorylation of the FOXP1 triple (3A) mutant when compared with the phosphorylation of wild-type FOXP1 (Fig. 4E). After testing each mutant in turn, we detected that the FOXP1 S396A mutant, but not the T236A mutant or S440A mutant, exhibited compromised phosphorylation (Figs. 4F and EV4C,D). We thus conclude that CHK1-mediated phosphorylation of FOXP1 occurs at S396.

Next, we generated an antibody that can specifically recognize the phosphorylation of FOXP1 at S396 so that we could examine the phosphorylation of enriched FLAG-tagged FOXP1 from HEK293T cells (Fig. EV4E). Using our antibody, we detected an elevated FOXP1 phosphorylation signal in response to replication stress, which was inhibited in the presence of CHK1 inhibitors Rabusertib or UCN-01, as well as when using the phospho-defective FOXP1 S396A mutant (Fig. 4G). We also used a phospho-mimic S396D mutant, and saw that in contrast to the phospho-deficient S396A mutant, this modified version of FOXP1 exhibited compromised O-GlcNAcylation, perhaps due to a reduced interaction between OGT and FOXP1 S396D (Figs. 4H,I and EV4F). Under conditions of replication stress, the FOXP1 S396A mutant, which failed to be phosphorylated by CHK1, exhibited a notably compromised interaction with ATR when compared to wild-type FOXP1 or its S396D mutant (Fig. 4J). Moreover, we examined the ATR activation in wild-type HEK293 cells and the cell lines with knock-in S396A or S396D mutation of FOXP1, and observed compromised ATR activation in the cell line expressing FOXP1 S396A mutant (Fig. EV4G). To explore whether FOXP1

phosphorylation influences its binding to ATR when O-GlcNAcylation is absent, we purified GST-tagged FOXP1 without O-GlcNAcylation from *E. coli* and performed a CHK1-mediated FOXP1 phosphorylation assay. We then conducted a GST pull-down experiment using unmodified and phosphorylated FOXP1. The results showed similar ATR levels interacting with S396-phosphorylated and unmodified FOXP1 (Fig. EV4H). Taken together, we consider that CHK1-mediated FOXP1 phosphorylation promotes its interaction with ATR by antagonizing its O-GlcNAcylation.

## FOXP1 pathogenic mutations impede FOXP1 function during the replication stress response

A FOXP1 deficiency is intimately related with a series of physiological disorders including developmental retardation and cancer (Gao et al, 2023; Rappold et al, 1993), the pathogenesis of which correlate with dysregulation of the replication stress response (Zeman and Cimprich, 2014). Most reported pathogenic FOXP1 mutations are found in the forkhead domain encompassing AA 465–555 (Siper et al, 2017). The R465 and R514 residues are recurrently mutated, with R465G and R514C mutations detected in patients with FOXP1 syndrome, with a characterized clinical symptom of developmental retardation (Hua et al, 2021; Siper et al, 2017; Sollis et al, 2016); and R465T, R514C, and R514H mutations in a broad spectrum of tumor tissues as indicated in TCGA database. We first performed an immunofluorescence experiment to examine the distribution of FOXP1 R465G, R514C, R465T, and R514H mutants in cells, and found none of these mutations affected the FOXP1 nuclear localization (Fig. EV5A). We did find, however, that FOXP1 chromatin loading during replication stress was compromised by each of the R465G, R514C, R465T, or R514H mutations (Fig. EV5B).

Given these findings, and that the FOXP1 forkhead domain is responsible for its binding with RPA–ssDNA (Figs. 2B and EV2C–F), we wondered whether these four mutants of FOXP1 would affect its function in the replication stress response. We first examined the binding of the FOXP1 mutants with ssDNA in vitro. Recombinant GST-tagged FOXP1 R465G, R514C, and R514H mutants showed markedly compromised affinity with ssDNA, while the R465T mutant only seemed to slightly decrease FOXP1's affinity for ssDNA (Fig. 5A). We also tested the interaction between RPA and these four FOXP1

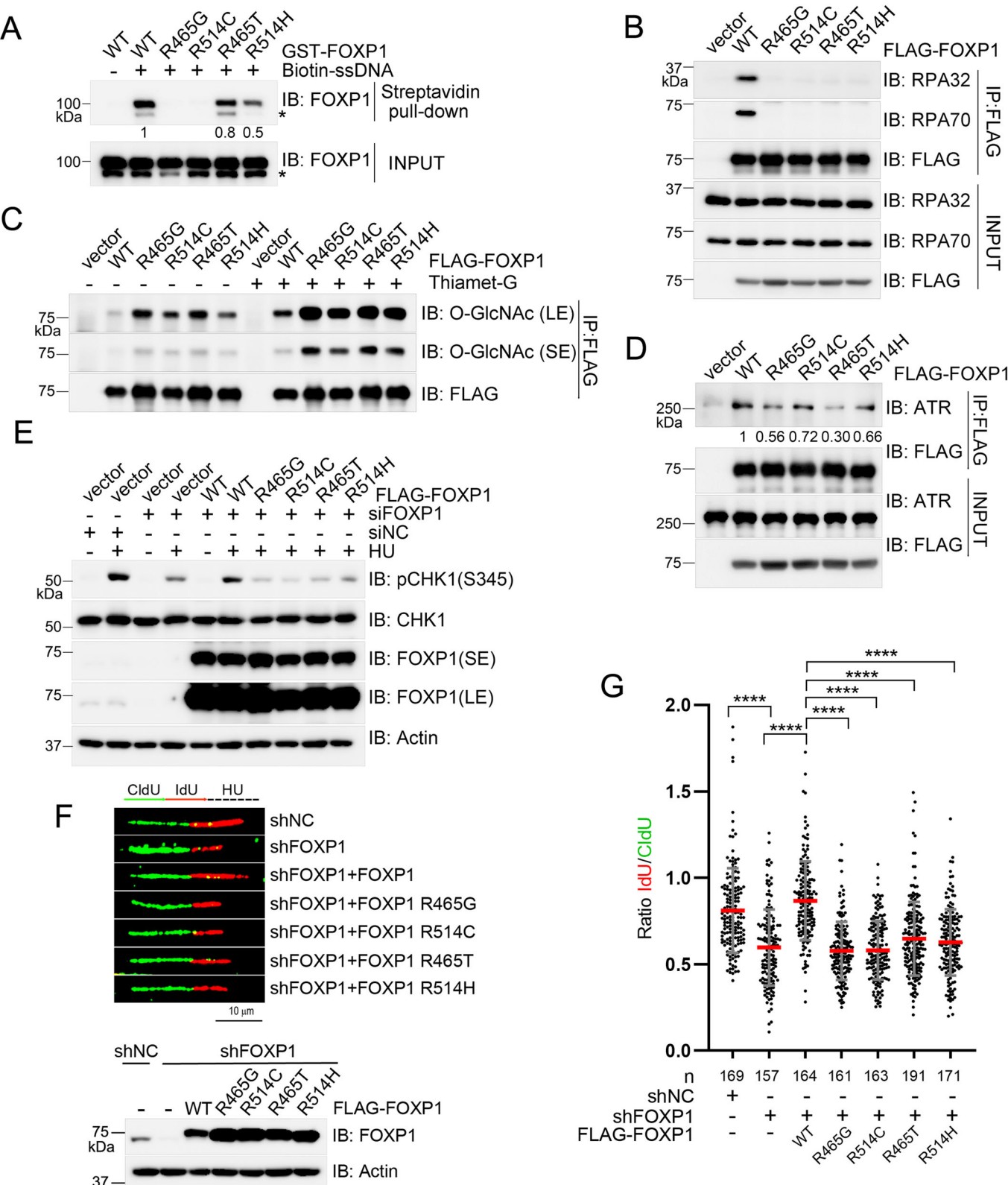

**Figure 5.  Pathogenic FOXP1 mutations compromise its function during conditions of replication stress.**

(A) Biotin-labeled ssDNA conjugated on streptavidin magnetic beads were incubated with GST-tagged FOXP1 or its mutants purified from *E. coli*. Streptavidin bead-bound FOXP1 was detected via immunoblotting using the indicated antibodies. * degraded GST-FOXP1. (B) HEK293T cells transfected with wild-type FLAG-FOXP1 or its pathogenic mutants were subjected to immunoprecipitation and immunoblotting using the indicated antibodies. (C) HEK293T cells transfected with wild-type FLAG-FOXP1 or its pathogenic mutants, preincubated with or without Thiamet-G, were subjected to immunoprecipitation and immunoblotting with the indicated antibodies. (D) HEK293T cells transfected with wild-type FLAG-FOXP1 or its pathogenic mutants were subjected to immunoprecipitation and immunoblotting using the indicated antibodies. (E) FOXP1 knockdown HEK293T cells transfected with wild-type FLAG-FOXP1 or its pathogenic mutants were treated with 2 mM HU for 1 h before being lysed for immunoblotting with the indicated antibodies. (F) Upper panel: schematic of the DNA fiber assay examining stalled replication fork stability. Middle panel: representative images of CldU and IdU replication tracks. Lower panel: FOXP1 levels in different H1975 cells. (G) Statistical analysis of the IdU/CldU ratio of DNA fibers (from F); the mean IdU/CldU ratio (red line) ±SD is shown. *n*, DNA fiber number, ****$P$ < 0.0001, $P$ values were calculated by one-way ANOVA, followed by Kruskal–Wallis test. $P$ value: shNC vs shFOXP1, 4.15e-015; shFOXP1 vs shFOXP1+WT, 1.93e-024; shFOXP1+WT vs shFOXP1 + R465G, 8.49e-030; shFOXP1+WT vs shFOXP1 + R514C, 4.98e-028; shFOXP1+WT vs shFOXP1 + R465T, 3.76e-018; shFOXP1+WT vs shFOXP1 + R514H, 3.69e-019. Source data are available online for this figure.

mutants; here, all four mutations compromised FOXP1's interaction with RPA (Fig. 5B). As a reminder, the mapping experiment of FOXP1 and OGT interaction showed that deletion of AA 465–555 promoted its interaction with OGT (Fig. 3D) and ensued O-GlcNAcylation (Fig. EV5C), but compromised the interaction with ATR (Fig. EV5D). We therefore wondered whether these pathologic mutants of FOXP1 led to aberrant O-GlcNAcylation and ATR interaction, and thereby dysregulated the replication stress response. Indeed, we detected that the R465G, R514C, R465T, and R514H FOXP1 mutants exhibited increased O-GlcNAcylation levels and a decreased interaction with ATR (Fig. 5C,D). Consistently, complementation of wild-type FOXP1, but not these four pathogenic mutants, promoted ATR activation and S phase arrest upon replication stress in FOXP1-deficient (siRNA-treated) HEK293 cells (Figs. 5E and EV5E). Moreover, results of a DNA fiber assay showed that these pathogenic FOXP1 mutations had a detrimental effect on stalled replication fork stability, as indicated by the decreased IdU/CldU ratio (Figs. 5F,G and EV5F). Collectively, these data suggest that FOXP1 R465 and R514 mutations negatively impact on ATR activation and stalled replication fork stabilization, which might partially underlie the pathogenesis of related diseases.

## Discussion

Proper and timely responses to replication stresses are essential for the survival and physiological functions of cells and organisms (Zeman and Cimprich, 2014). As a notable part of the replication stress response network, ATR signaling is pivotal in protecting the stalled replication forks and maintaining the genomic stability (Saldivar et al, 2017). Activation of ATR kinase is under regulation by complicated mechanisms with the interplays between different PTMs, while the precise details are not fully understood. In our findings, we identified that FOXP1 served as a scaffold protein to facilitate the recruitment of ATR and subsequent ATR activation, via directly binding to both RPA-coated ssDNA and the ATR–ATRIP complex, while the loading of TOPBP1 and ETAA1 are not affected in FOXP1-depleted cells (Fig. 1H). Additionally, FOXP1 does not contain a clear ATR-activating domain like those found in TopBP1 and ETAA1, as previously reported (Bass et al, 2016). Meanwhile, our data support that FOXP1 O-GlcNAcylation represses its interaction with ATR, and FOXP1 CHK1-mediated phosphorylation at S396 antagonizes its O-GlcNAcylation under conditions of replication stress. Thus, the interplay between these

two PTMs of FOXP1 forms a feed-forward loop in activating ATR signaling (Fig. 6).

*FOXP1* was first identified as a potential tumor suppressor gene mapped on chromosome 3p14.1 (Banham et al, 2001). However, emerging data support that FOXP1 has dual functions as both a tumor suppressor and oncoprotein in the context of different cancers (Gao et al, 2023). For example, FOXP1 is a tumor suppressor in lung cancer, pancreatic cancer, cholangiocarcinoma, prostate cancer, breast cancer, and neuroblastoma (Ackermann et al, 2014; Fox et al, 2004; Sheng et al, 2019; Takayama et al, 2014; Tang et al, 2024; Wang et al, 2023), but an oncoprotein in DLBCL and ovarian cancer, and sometimes also exhibit oncoprotein functions in breast cancer and prostate cancer (Chen et al, 2023; Chiang et al, 2017; Choi et al, 2016; Takayama et al, 2008). While FOXP1 positively regulates the expression level of GINS1 in DLBCL and promotes DLBCL proliferation (Chen et al, 2023), we found that the GINS1 level was not reduced in FOXP1-deficient H1975 cells or HEK293T cells, indicating the different functions of FOXP1 in lymphoma and solid tumors. In addition, FOXP1 is reported to be an estrogen-inducible and androgen-inducible transcription factor (Shigekawa et al, 2011; Takayama et al, 2008) —a factor that may be correlated with its oncogenic function in hormone-sensitive breast cancer, ovarian cancer, and prostate cancer.

Moreover, FOXP1 is a multisystemic regulator, necessary for normal nervous system development, lung development, cardio-vascular development, naive T-cell quiescence, and B-cell development (Anderson et al, 2020; Dekker et al, 2019; Feng et al, 2011; Patzelt et al, 2018; Usui et al, 2017; Wei et al, 2016; Zhang et al, 2010). FOXP1 haploinsufficiency is implicated in the pathogenesis of FOXP1 syndrome, which is clinically characterized by developmental disorders (Rappold et al, 1993). The pathogenic mutations identified in patients with FOXP1 syndrome mainly locate in its forkhead domain (Meerschaut et al, 2017; Siper et al, 2017), and this domain also contains multiple mutations in a broad spectrum of cancers as indicated in the TCGA database. The FOXP1 forkhead domain confers its DNA binding capacity and transcription factor activity (Wang et al, 2003). The higher affinity of FOXP1 forkhead domain for ssDNA rather than dsDNA in vitro, supported by our data (Fig. 2A), indicates the neglected functions of FOXP1 at specific DNA structures. As a transcription factor, FOXP1 has a preference for binding the consensus motif GTAAACA (Gabut et al, 2011; Li et al, 2004), although FOXP1 has a preference for binding dsDNA containing the GTAAACA

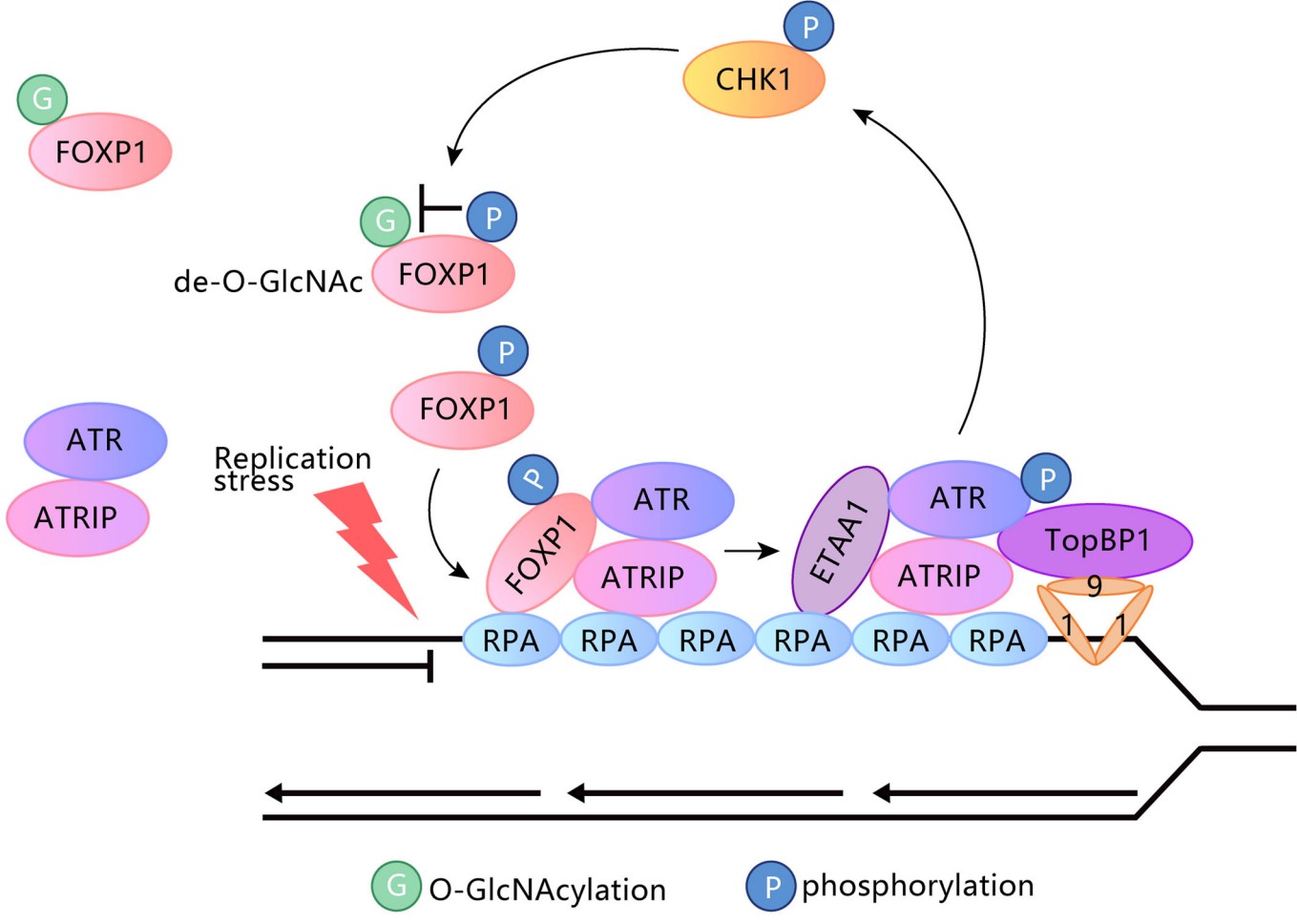

**Figure 6. Working model of FOXP1 function during the replication stress response.**

FOXP1 served as a scaffold protein to facilitate the recruitment of ATR and subsequent ATR activation, via directly binding to both RPA-coated ssDNA and the ATR–ATRIP complex. FOXP1 O-GlcNAcylation represses its interaction with ATR, and FOXP1 CHK1-mediated phosphorylation at S396 antagonizes its O-GlcNAcylation under conditions of replication stress.

consensus (Fig. EV2A), its affinity for ssDNA was similar, regardless of whether the sequence includes the GTAAACA motif (Fig. EV2B). This suggests that FOXP1 regulates transcription at specific DNA regions but protects stalled replication forks more broadly. Meanwhile, in our findings, the forkhead domain of FOXP1 also mediates its physical binding to the RPA complex and ATR–ATRIP complex (Fig. EV5G), and confers its scaffold function beyond the transcription regulation. However, whether the capacity of the forkhead domain to bind to other proteins and its preference for binding to ssDNA is restricted to FOXP1, or other FOXP subfamily members, or even the whole FOX family members are not clear and need further elucidation. In addition, except for the scaffold function of FOXP1, as an RNA polymerase II-dependent transcription repressor (Gaudet et al, 2011), whether FOXP1 also exerts its transcription regulatory activity at replication forks when encountering the transcription machinery worth further study. In our findings, the uncovered function of FOXP1 as a scaffold protein in facilitating ATR activation under conditions of replication stress, may partially explain the pathogenesis of

FOXP1-deficiency-related tumorigenesis and developmental disorder and potentially serves as a chemotherapeutic target.

## Methods

**Reagents and tools table**

| Reagent/resource | Reference or source | Identifier or catalog number |
|---|---|---|
| **Experimental models** | | |
| HEK293T cells (*H. sapiens*) | ATCC | CRL-3216 |
| NCI-H1975 cells (*H. sapiens*) | ATCC | CRL-5908 |
| HEK293 cells (*H. sapiens*) | ATCC | CRL-1573 |
| **Recombinant DNA** | | |
| pcDNA3.0-FLAG-FOXP1 | This study | N/A |
| pcDNA3.0-FLAG-FOXP1 R465G | This study | N/A |

| Reagent/resource | Reference or source | Identifier or catalog number |
|---|---|---|
| pcDNA3.0-FLAG-FOXP1 R514C | This study | N/A |
| pcDNA3.0-FLAG-FOXP1 R465T | This study | N/A |
| pcDNA3.0-FLAG-FOXP1 R514H | This study | N/A |
| pcDNA3.0-FLAG-FOXP1 S396A | This study | N/A |
| pcDNA3.0-FLAG-FOXP1 S396D | This study | N/A |
| pcDNA3.0-FLAG-FOXP1Δ2-230 | This study | N/A |
| pcDNA3.0-FLAG-FOXP1Δ231-369 | This study | N/A |
| pcDNA3.0-FLAG-FOXP1Δ370-390 | This study | N/A |
| pcDNA3.0-FLAG-FOXP1Δ391-410 | This study | N/A |
| pcDNA3.0-FLAG-FOXP1Δ411-430 | This study | N/A |
| pcDNA3.0-FLAG-FOXP1Δ431–450 | This study | N/A |
| pcDNA3.0-FLAG-FOXP1Δ451–464 | This study | N/A |
| pcDNA3.0-FLAG-FOXP1Δ465–555 | This study | N/A |
| pcDNA3.0-FLAG-FOXP1Δ556–610 | This study | N/A |
| pcDNA3.0-FLAG-FOXP1Δ611-677 | This study | N/A |
| pcDNA3.0-FLAG-FOXP1Δ556–570 | This study | N/A |
| pcDNA3.0-FLAG-FOXP1Δ571–590 | This study | N/A |
| pcDNA3.0-FLAG-FOXP1Δ591–610 | This study | N/A |
| pLKO.1-shFOXP1 | This study | N/A |
| plenti-Blast-FOXP1 | This study | N/A |
| pGEX-4T-1-FOXP1 | This study | N/A |
| pGEX-4T-1-FOXP1 S396A | This study | N/A |
| pGEX-4T-1-FOXP1 T236A | This study | N/A |
| pGEX-4T-1-FOXP1 S440A | This study | N/A |
| pGEX-4T-1-FOXP1 T236A/S396A/S440A | This study | N/A |
| pGEX-4T-1-FOXP1 Δ431–450 | This study | N/A |
| pGEX-4T-1-FOXP1 Δ451–464 | This study | N/A |
| pGEX-4T-1-FOXP1 Δ465–555 | This study | N/A |
| pet28a-OGT | This study | N/A |
| pet28a-RPA32 | This study | N/A |
| pet28a-RPA70 | This study | N/A |
| pEGFP-C1-OGT | Dr. Qiang Chen (Wuhan University) | N/A |
| pS-FLAG-SBP-OGT | Dr. Qiang Chen (Wuhan University) | N/A |
| pGEX-6P-1-ATRIP | Dr. Wei-Guo Zhu (Shenzhen University) | N/A |
| **Antibodies** | | |
| Rabbit anti-ATR polyclonal antibody | Bethyl Laboratories | Cat # A300-138A |
| Rabbit anti-RPA32 polyclonal antibody | Bethyl Laboratories | Cat # A300-244A |
| Rabbit anti-RPA70 polyclonal antibody | Bethyl Laboratories | Cat # A300-241A |

| Reagent/resource | Reference or source | Identifier or catalog number |
|---|---|---|
| Rabbit anti-CHK1 polyclonal antibody | Bethyl Laboratories | Cat # A300-298A |
| Mouse anti-CHK1 monoclonal antibody | Santa Cruz Biotechnology | Cat # sc-8408 |
| Rabbit anti-GINS1 polyclonal antibody | Bethyl Laboratories | Cat # A304-170A |
| Rabbit anti-ATRIP polyclonal antibody | Cell Signaling Technology | Cat # 2737 |
| Rabbit anti-ATRIP polyclonal antibody | ABclonal Technology | Cat # A5041 |
| Rabbit anti-FOXP1 polyclonal antibody | ABclonal Technology | Cat # A12685 |
| Rabbit anti-FOXP1 monoclonal antibody | PTM BIO | Cat # A5666 |
| Rabbit anti- pFOXP1 (S396) polyclonal antibody | This study (by PTM BIO) | N/A |
| Rabbit anti-Actin monoclonal antibody | ABclonal Technology | Cat # AC026 |
| Rabbit anti-pCHK1(S345) monoclonal antibody | Cell Signaling Technology | Cat # 2348 |
| Rabbit anti-pCHK1(S296) monoclonal antibody | Abcam | Cat # ab79758 |
| Rabbit anti-TopBP1 monoclonal antibody | Cell Signaling Technology | Cat # 14342 |
| Mouse anti-O-GlcNAc monoclonal antibody (CTD110.6) | Cell Signaling Technology | Cat # 9875 |
| Mouse anti-O-GlcNAc monoclonal antibody (RL2) | Abcam | Cat # ab2739 |
| Mouse anti-GFP monoclonal antibody | Santa Cruz Biotechnology | Cat # sc-9996 |
| Mouse anti-Biotin monoclonal antibody | Santa Cruz Biotechnology | Cat # sc-53179 |
| Mouse anti-FLAG monoclonal antibody | Sigma-Aldrich | Cat # F1804 |
| Rabbit anti-HA polyclonal antibody | Proteintech Group | Cat # 51064-2-AP |
| Rabbit anti-PCNA polyclonal antibody | Proteintech Group | Cat # 10205-2-AP |
| Rabbit anti-OGT polyclonal antibody | Proteintech Group | Cat # 11576-2-AP |
| Rabbit anti-ETAA1 monoclonal antibody | Abcam | Cat # ab197017 |
| Rabbit anti-thiophosphate ester monoclonal antibody | Abcam | Cat # ab92570 |
| Rabbit anti-H3 polyclonal antibody | Abcam | Cat # ab18521 |
| Mouse anti-GST monoclonal antibody | MBL Life Science | Cat # M209-3 |
| Mouse anti-His monoclonal antibody | MBL Life Science | Cat # D291-3 |
| Rat anti-BrdU monoclonal antibody | Abcam | Cat # ab6326 |
| Mouse anti-BrdU monoclonal antibody | BD Biosciences | Cat # 347580 |
| Donkey anti-Mouse IgG (H + L) Secondary Antibody, Alexa Fluor™ 594 | Thermo Fisher Scientific | Cat # A-21203 |

| Reagent/resource | Reference or source | Identifier or catalog number |
|---|---|---|
| Alexa Fluor® 488 AffiniPure™ Donkey Anti-Rat IgG (H + L) | Jackson ImmunoResearch Laboratories | Cat # 712-546-150 |
| Peroxidase AffiniPure™ Goat Anti-Mouse IgG (H + L) | Jackson ImmunoResearch Laboratories | Cat # 115-035-166 |
| Peroxidase AffiniPure™ Donkey Anti-Rabbit IgG (H + L) | Jackson ImmunoResearch Laboratories | Cat # 711-035-152 |
| Peroxidase IgG Fraction Monoclonal Mouse Anti-Rabbit IgG, light chain-specific | Jackson ImmunoResearch Laboratories | Cat # 211-032-171 |
| Peroxidase AffiniPure™ Goat Anti-Mouse IgG, light chain-specific | Jackson ImmunoResearch Laboratories | Cat # 115-035-174 |
| AffiniPure™ Goat Anti-Mouse IgM, μ chain-specific | Jackson ImmunoResearch Laboratories | Cat # 115-005-020 |
| **Oligonucleotides and other sequence-based reagents** | | |
| PCR primers | This study | Table EV1 |
| siRNA sequence | This study | Table EV1 |
| shRNA sequence | This study | Table EV1 |
| **Chemicals, enzymes, and other reagents** | | |
| Dulbecco's modified Eagle's medium | HyClone | Cat # SH30243.01 |
| Fetal bovine serum | ExCell Bio | Cat # FSP500 |
| Penicillin-streptomycin | HyClone | Cat # SV30010 |
| Protein A Sepharose™ CL-4B | Cytiva Lifesciences | Cat # 17078001 |
| Glutathione Sepharose™ 4B | Cytiva Lifesciences | Cat # 17075601 |
| HisSep Ni-NTA Agarose Resin 6FF | Yeasen Biotechnology | Cat # 20503ES50 |
| Anti-DYKDDDDK (Flag) Affinity Gel | Selleck | Cat # B23102 |
| Lipofectamine™ RNAiMAX Transfection Reagent | Thermo Fisher Scientific | Cat # 13778150 |
| Polyethylenimine Linear (PEI) MW40000 | Yeasen Biotechnology | Cat # 40816ES03 |
| Hydroxyurea | Selleck | Cat # S1896 |
| Thiamet-G | Selleck | Cat # S7213 |
| Rabusertib | Selleck | Cat # S2626 |
| UCN-01 | Sigma-Aldrich | Cat # 539644 |
| VE-822 | Selleck | Cat # S7102 |
| NU6027 | Selleck | Cat # S7114 |
| Protease inhibitor cocktail | TargetMol | Cat # C0001 |
| Phosphatase inhibitor cocktail | TargetMol | Cat # C0002 |
| CldU | Sigma-Aldrich | Cat # C6891 |
| IdU | Sigma-Aldrich | Cat # I7125 |
| EdU | Selleck | Cat # S1661 |
| BrdU | TargetMol | Cat # T6794 |

| Reagent/resource | Reference or source | Identifier or catalog number |
|---|---|---|
| CPT | Sigma-Aldrich | Cat # S1288 |
| p-Nitrobenzyl mesylate | Abcam | Cat # ab138910 |
| ATP-γ-S | Abcam | Cat # ab138911 |
| ATP-Na$_2$ | Beyotime Biotechnology | Cat # ST1092 |
| Recombinant Human Chk1 Protein (GST Tag) | SinoBiological | Cat # 10539-H09B |
| 2 × Phanta Flash Master Mix | Vazyme | Cat # P510 |
| pEASY®-Basic Seamless Cloning and Assembly Kit | TransGen Biotech | Cat # CU201 |
| T4 ligase | New England Biolabs | Cat # M0202 |
| Trans5α Chemically Competent Cell | TransGen Biotech | Cat # CD201 |
| Transetta(DE3) Chemically Competent Cell | TransGen Biotech | Cat # CD801 |
| Duolink In Situ Detection Reagents Red | Sigma-Aldrich | Cat # DUO92008 |
| Duolink In Situ PLA Probe Anti-Mouse PLUS | Sigma-Aldrich | Cat # DUO92001 |
| Duolink In Situ PLA Probe Anti-Rabbit MINUS | Sigma-Aldrich | Cat # DUO92005 |
| UDP-GalNAz | Glycogene | Cat # SN-1016 |
| DBCO-mPEG5000 | Confluore | Cat # BCDG-9 |
| **Software** | | |
| GraphPad Prism 8 | https://www.graphpad.com/ | |
| Image J | https://imagej.net | |
| FlowJo | https://www.flowjo.com/ | |

## Methods and protocols

### Cell culture and transfection

HEK293T, H1975, and HEK293 cell lines were obtained from the American Type Culture Collection. Cells were cultured with high-glucose Dulbecco's modified Eagle's medium (HyClone) supplemented with 10% fetal bovine serum (ExCell Bio) and penicillin-streptomycin (HyClone), at 37 °C in a humidified 5% CO$_2$ incubator. The short tandem repeat (STR) profiles of cell lines are shown in Table EV2. Cell transfection was performed using 1 mg/ml Polyethylenimine Linear (PEI, Yeasen Biotechnology) following the manufacturer's protocol.

### Plasmid constructs

Human full-length FOXP1 cloned into a pcDNA3.0 expression vector with a FLAG N-terminal epitope using 2 × Phanta Flash Master Mix (Vazyme) and a pEASY®-Basic Seamless Cloning and Assembly Kit (TransGen Biotech). S396A, S396D, R465G, R514C, R465T, and R514H mutants, and all the fragment deletion mutants of FOXP1 were PCR-amplified using primers containing the mutations. Wild-type FOXP1

and its mutants were also cloned into a lenti-Blast vector. shRNAs (forward oligo: CCGGGCATTGGATGGACTTGTTTCTCGAGAAA-CAAGTCCATCCAATGCTTTTTG; reverse oligo: AATTCAAAAAG-CATTGGATGGACTTGTTTCTCGAGAAACAAGTCCATCCAAT GC) targeting the 3′-UTR regions of *FOXP1* were cloned into a PLKO.1 vector. Full-length wild-type FOXP1 (human), single point mutations, or deletion mutations were cloned into a pGEX-4T-1 vector. Full-length OGT (human), RPA32 (human), and RPA70 (human) were cloned into a pet28a vector. GFP-OGT (human) and SFB-OGT (human) are kind gifts from Dr. Qiang Chen, Wuhan University; and GST-ATRIP (human) is a kind gift from Dr. Wei-Guo Zhu, Shenzhen University.

### RNA interference

The introduction of small interfering RNA (siRNA) into HEK293T, H1975, or HEK293 cells were carried out using RNAiMAX following the manufacturer's protocol. The siRNAs directed against FOXP1 were synthesized by RiboBio. The siRNA sequences were as follows: siNC: UUCUCCGAACGUGUCACGU; siFOXP1(CDS): CUGGUUCACACG AAUGUUU; siFOXP1 (3′-UTR): GCAUUGGAUGGACUUGUUU.

### FOXP1 mutant knock-in cell line

FOXP1 S396A and S396D knock-in cell lines were generated in HEK293 cells (by Ubigene). Knock-in cell lines were established using CRISPR-Cas9 genome-editing technology. The guide RNA for S396A (TCTCTCCAAGTCCGCATCGG AGG) and for S396D (CTGTGG AGAAGCCTCCGATG CGG) were designed. The mutations c.T118 6G(TCG > GCG, p.S396A) and c.T1186G c.C1187A c.G1188T(TCG > GAT, p.S396D) were introduced into exon15 by homology-directed repair. The monoclonal mutant cell line genome was confirmed by PCR identification (PF: GGCCTGATGGGCGATCAAAG, PR: GAG CTCAGATTAATTCTAGGGATC).

### Immunoprecipitation, pull-down assay, and immunoblotting

HEK293T cells transfected with the indicated plasmids were lysed in NETN buffer (100 mM NaCl, 1 mM EDTA, 20 mM Tris-HCl pH 8.0 and 0.5% NP-40) containing protease and phosphatase inhibitors for 30 min at 4 °C and then centrifuged. For endogenous immunoprecipitation, cell lysates were incubated with the indicated antibodies for 4 h at 4 °C and then incubated with protein-A beads for 1 h, followed by extensive washes with NETN buffer for 10 min × three times at 4 °C. For exogenous immunoprecipitation, cell lysates were incubated with Anti-FLAG Affinity Gel for 4 h, followed by extensive washes with NETN buffer at 4 °C. Bead-bound proteins were denatured in 2× sample buffer (62.5 mM Tris-HCl pH 6.8, 2% SDS, 20 mM DTT and 10% glycerol) at 100 °C for 5 min, and then resolved by SDS-PAGE and examined by immunoblotting with the indicated antibodies.

For His pull-down assays, bacterially purified GST-FOXP1 and His-RPA32 or His-RPA70 immobilized on Ni-NTA Agarose were incubated for 4 h at 4 °C. For GST pull-down assays, bacterially purified His-RPA32 or His-RPA70 and GST-FOXP1 or its deletion mutants immobilized on Glutathione Sepharose 4B beads were incubated for 4 h at 4 °C. Then, bead-bound proteins were denatured in 2× sample buffer and examined by immunoblotting.

For immunoblotting, samples were separated by SDS-PAGE and transferred to the PVDF blotting membrane (Cytiva) before blocking with skim milk and then blotting with the indicated antibodies. Primary antibodies were incubated overnight at 4 °C while secondary antibodies incubated for 2 h at room temperature. A Super Signal

West Femto Substrate kit was used to visualize proteins after processing membranes using an Amersham Imager 600 system.

### Immunofluorescence

H1975 cells expressing FLAG-tagged FOXP1 or the mutants grown on the coverslips were fixed with 4% paraformaldehyde (PFA) for 20 min. The cells were washed twice with PBS before being permeabilized with 0.5% Triton X-100 (in PBS) for 5 min. The cells were blocked with 2% BSA for 30 min before being incubated with primary and then secondary antibodies for 1 h each at 37 °C. The nuclei were stained with DAPI for 2 min and then images were captured under a DragonFly confocal imaging system (Andor).

### Proximity ligation assay

H1975 cells grown on the coverslips were labeled with 10 μM EdU for 15 mins at 37 °C before treated with or without 2 mM HU for 1 h. The cells were then fixed with 4% PFA for 20 min, and washed twice with PBS before permeabilized with 0.5% Triton X-100 (in PBS) for 5 min and blocked with 2% BSA for 30 min. A click chemistry reaction was performed to conjugate EdU with biotin-azide. Next, the cells were incubated with biotin-specific and FOXP1-specific antibodies at 4 °C overnight. The subsequent procedures were carried out according to the manufacturer's instructions of the Duolink In Situ Red Starter kit (Sigma-Aldrich). Images were captured using a DragonFly confocal imaging system (Andor).

### iPOND

Isolation of proteins on nascent DNA (iPOND) was performed as previously described with some modifications (Sirbu et al, 2012). In brief, cells were labeled with 10 μM EdU for 15 min before treatment or not with HU for 1 h. The cells were then fixed with 1% formaldehyde for 20 min at RT, followed by quenching with 1.25 M glycine for 5 min. Next, the cells were collected and permeated in 0.25% Triton X-100 for 30 min on ice. Before the next step, a click chemistry reaction was prepared to conjugate biotin to EdU, with 10 μM biotin-azide for 1.5 h at 4 °C. After rinsing with PBS, cells were resuspended in lysis buffer (1% SDS in 50 mM Tris-HCl, PH 8.0). Then, the cells were sonicated for 10 min at high intensity to obtain 100- to 300-bp fragments. Streptavidin-conjugated Dynabeads M-280 (Invitrogen) were added to enrich EdU labeled fragments and incubated at 4 °C overnight, and the bound proteins were analyzed by immunoblotting.

### DNA fiber assay

H1975 cells with the indicated FOXP1 background were sequentially labeled with 40 μM CldU and 100 μM IdU for 0.5 h each at 37 °C to detect DNA replication or followed by treatment with 5 mM HU for 4 h to detect stalled replication fork stability. Then, the cells were dissociated by trypsinization and mixed with unlabeled cells before performing a DNA fiber assay, as described previously (Xu et al, 2017).

### Cell cycle distribution analysis

Cells were labeled with 10 μM BrdU for 30 min, harvested by trypsinization and washed twice with cold PBS before being fixed with ice-cold 70% ethanol for 16 h. The cells were then washed twice with 1% BSA and permeabilized with 0.5% Triton X-100 for

20 min, denatured with 2 M HCl for 30 min, and neutralized with 0.1 M $Na_2B_4O_7$ for 10 min. After being washed twice with 1% BSA, the cells were incubated with an anti-BrdU (BD347580) antibody for 1 h at 37 °C and washed with PBST (0.1% Tween-20) for three times. FITC-conjugated secondary antibody incubation was performed for 1 h at 37 °C. After washing with PBST three times, cells were resuspended in propidium iodide/RNase and incubated for 30 min before the cell cycle distribution was analyzed by flow cytometry.

### Chromatin fractionation

Preparation of the chromatin fraction was performed as described previously with some modifications (Kannouche et al, 2004). In brief, cells were rinsed with cold PBS, and incubated for 10 min on ice in CSK buffer (10 mM Pipes pH 6.8, 100 mM NaCl, 300 mM sucrose, 3 mM MgCl2, 1 mM EGTA, 0.2% Triton X-100) with protease inhibitor. After rinsing with cold PBS, the cells were collected and centrifuged. The soluble fraction was collected, and the pellets were then incubated with lysis buffer (50 mM HEPES pH 7.5, 50 mM NaCl, 0.05% SDS, 2 mM MgCl2, 10% glycerol, 0.1% Triton X-100) containing benzonase overnight at 4 °C before centrifugation to collect the supernatant. For immunoprecipitation, both the soluble fraction and chromatin fraction (diluted ten times with NETN buffer) were subjected to immunoprecipitation using Anti-FLAG Affinity Gel.

### In vitro phosphorylation assay

For in vitro phosphorylation with ATP-γ-S, GST-CHK1 and GST-FOXP1 (or its mutants) were incubated in reaction mixture (50 mM Tris-HCl pH 7.5, 10 mM MgCl₂, 20 μM ATP-γ-S, 1 mM DTT, and 50 μM $Na_3VO_4$) at 37 °C for 30 min. Then the thiophosphate species were alkylated with p-nitrobenzyl mesylate (PNBM) for 2 h at room temperature before being denatured in sample buffer and examined by immunoblotting.

For in vitro phosphorylation of FOXP1 by CHK1 with ATP. His-CHK1 and GST-FOXP1 (or its mutants) were incubated in reaction mixture (50 mM Tris-HCl pH 7.5, 10 mM MgCl₂, 20 μM ATP, 1 mM DTT, and 50 μM $Na_3VO_4$) at 37 °C for 60 min.

### In vitro DNA binding assay

Biotin-labeled single-stranded DNA (TGCAGCTGGCACGACAG GTTTTAATGAATCGGCCAACGCGCGGGGAGAGGCGGTTTG CGTATTGGGCGCTCTTCCGCTTCGCAGCGAGTC-biotin-3′) was incubated with Dynabeads M-280 Streptavidin for 30 min at room temperature, washed with NETN buffer three times, further incubated with His-RPA70/His-RPA32 and washed with NETN buffer three times. Then GST-FOXP1 was incubated with Dynabeads M-280 Streptavidin-conjugated naked ssDNA or RPA–ssDNA for 30 min at room temperature. Bead-bound proteins were denatured in 2× sample buffer and examined by immunoblotting.

For specific sequence DNA binding assay, AACCTGTCGTGCCA GCTGCA-biotin-3′ and AACCTGTC**GTAAACA**CTGCA-biotin-3′, in which the bold nucleotide sequence represents the binding motif for FOXP1, or their complemented dsDNA were used for the experiments as described before.

### O-GlcNAcylation stoichiometry analysis

Immunoprecipitated beads with FLAG-tagged FOXP1 were labeled with GalNAz as previously reported (Thompson et al, 2018),

followed by conjugation with 1 mM DBCO-mPEG5000 for 3 h at 30 °C. Control experiments in the absence of Gal-T1 (Y289L) were carried out in parallel. After conjugation, the beads were mixed with 2×SDS sample buffer, denatured at 100 °C for 5 min and analyzed by immunoblotting with FLAG-specific antibody.

### Statistical analysis

Statistical analyses were performed in GraphPad Prism 8. The data of two groups with normal distribution were analyzed using unpaired two-tailed $t$ test. Data from multiple groups were analyzed using one-way ANOVA, followed by Dunnett's test for normally distributed data and Kruskal–Wallis test for data without a normal distribution. Data from multiple groups with two variables were analyzed using two-way ANOVA, followed by Sidak's test. In all cases, $P < 0.05$ was considered to indicate a statistical significance of difference.

## Data availability

This study includes no data deposited in external repositories.

The source data of this paper are collected in the following database record: biostudies:S-SCDT-10_1038-S44318-024-00323-x.

## Peer review information

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

## Acknowledgements

The authors would like to thank all members of the Xu laboratory for their help and useful discussions. We thank Dr. Qiang Chen (Wuhan University) and Dr. Wei-Guo Zhu (Shenzhen University) for providing plasmids. This work was supported by the National Natural Science Foundation of China (NSFC) (Grant Nos. 32200585, 32090031, 32371358), National Key R&D Program of China (2022YFA1302804), the Shenzhen Science and Technology Innovation Commission (Grant Nos. JCYJ20220818095616035, JCYJ20220818095605011), the Shenzhen Key Medical Discipline Construction Fund & Sanming Project of Medicine in Shenzhen (SZSM202111002), and the Program for Youzuzhikeyan of Shenzhen University.

## Author contributions

**Xuefei Zhu**: Conceptualization; Data curation; Formal analysis; Funding acquisition; Investigation; Methodology; Writing—original draft; Writing—review and editing. **Congwen Gao**: Data curation; Formal analysis; Validation; Investigation; Methodology. **Bin Peng**: Conceptualization; Data curation; Funding acquisition; Validation; Investigation; Methodology. **Jingwei Xue**: Data curation; Formal analysis. **Donghui Xia**: Data curation; Formal analysis; Validation; Methodology. **Liu Yang**: Validation. **Jiexiang Zhang**: Validation. **Xinrui Gao**: Data curation. **Yilin Hu**: Data curation; Methodology. **Shixian Lin**: Data curation; Methodology. **Peng Gong**: Conceptualization; Supervision; Writing—review and editing. **Xingzhi Xu**: Conceptualization; Data curation; Supervision; Funding acquisition; Validation; Project administration; Writing—review and editing.

Source data underlying figure panels in this paper may have individual authorship assigned. Where available, figure panel/source data authorship is listed in the following database record: biostudies:S-SCDT-10_1038-S44318-024-00323-x.

## Disclosure and competing interests statement

The authors declare no competing interests.

# Expanded View Figures

**Figure EV1. FOXP1 promotes ATR activation.**

(A) Coomassie blue staining of immunoprecipitates enriched with ATR-specific antibody. (B) His-tagged FOXP1 was incubated with GST-tagged ATRIP followed by GST pull-down assay. Proteins bound onto Glutathione beads were detected via immunoblotting using the indicated antibodies. (C) H1975 cells transfected with negative control siRNA or different siRNA targeting FOXP1 were treated with HU for 1 h before the whole cell lysates were harvested for immunoblotting with the indicated antibodies. (D) Statistical analysis of the IdU/CldU ratio mean in Fig. 1E, mean (biological replicates, $n = 3$) ± SD is shown. ***$P < 0.001$; ns, no significance; *P* values were calculated by one-way ANOVA, followed by Dunnett's test. *P* value: siNC vs siFOXP1-1, 0.0004; siNC vs siFOXP1-2, 0.0001; siFOXP1-1 vs siFOXP1-2, 0.9510. (E) HEK293T or H1975 cells transfected with siNC or siFOXP1 were pulse labeled with 10 µM BrdU for 30 min, followed by BrdU and PI staining and flow cytometric analysis. The percentage represents the BrdU-positive cells. (F) HEK293 cells transfected with negative control siRNA or different siRNA targeting FOXP1 were incubated with 100 nM CPT for 8 h or left untreated before harvested for PI staining and flow cytometric analysis. The percentage of S phase population was analyzed, mean (biological replicates, $n = 3$) ± SD is shown. ****$P < 0.0001$, *P* values were calculated by two-way ANOVA, followed by Sidak's test. *P* value: siNC vs siFOXP1-1, 6.40e-006; siNC vs siFOXP1-2, 1.02e-006. (G) The whole cell lysates of HEK293T or H1975 cells transfected with siNC or siFOXP1 were harvested for immunoblotting with the indicated antibodies. (H) H1975 cells with different FOXP1 levels were subjected to sequential labeling with CldU and IdU for 30 min each, followed by DNA fiber assay. Left: representative images of CldU and IdU replication tracks. Middle: the IdU tract length was analyzed, IdU length mean (red line) ±SD is shown in scatter plot, n, DNA fiber number; ns, no significance; *P* values were calculated by one-way ANOVA, followed by Kruskal–Wallis test. *P* value: siNC vs siFOXP1-1, $P > 0.9999$; siNC vs siFOXP1-2, $P > 0.9999$; siFOXP1-1 vs siFOXP1-2, $P > 0.9999$. Right: Mean IdU length ± SD (biological replicates, $n = 3$) was also analyzed and shown in column, ns, no significance, *P* values were calculated by one-way ANOVA, followed by Dunnett's test. *P* value: siNC vs siFOXP1-1, 0.9898; siNC vs siFOXP1-2, 0.7224; siFOXP1-1 vs siFOXP1-2, 0.6490. (I) HEK293T cells transfected with FLAG-FOXP1 treated with 2 mM HU for 1 h were subjected to chromatin fractionation. Both the soluble and chromatin fractions were subjected to immunoprecipitation using a FLAG-specific antibody. Source data are available online for this figure.

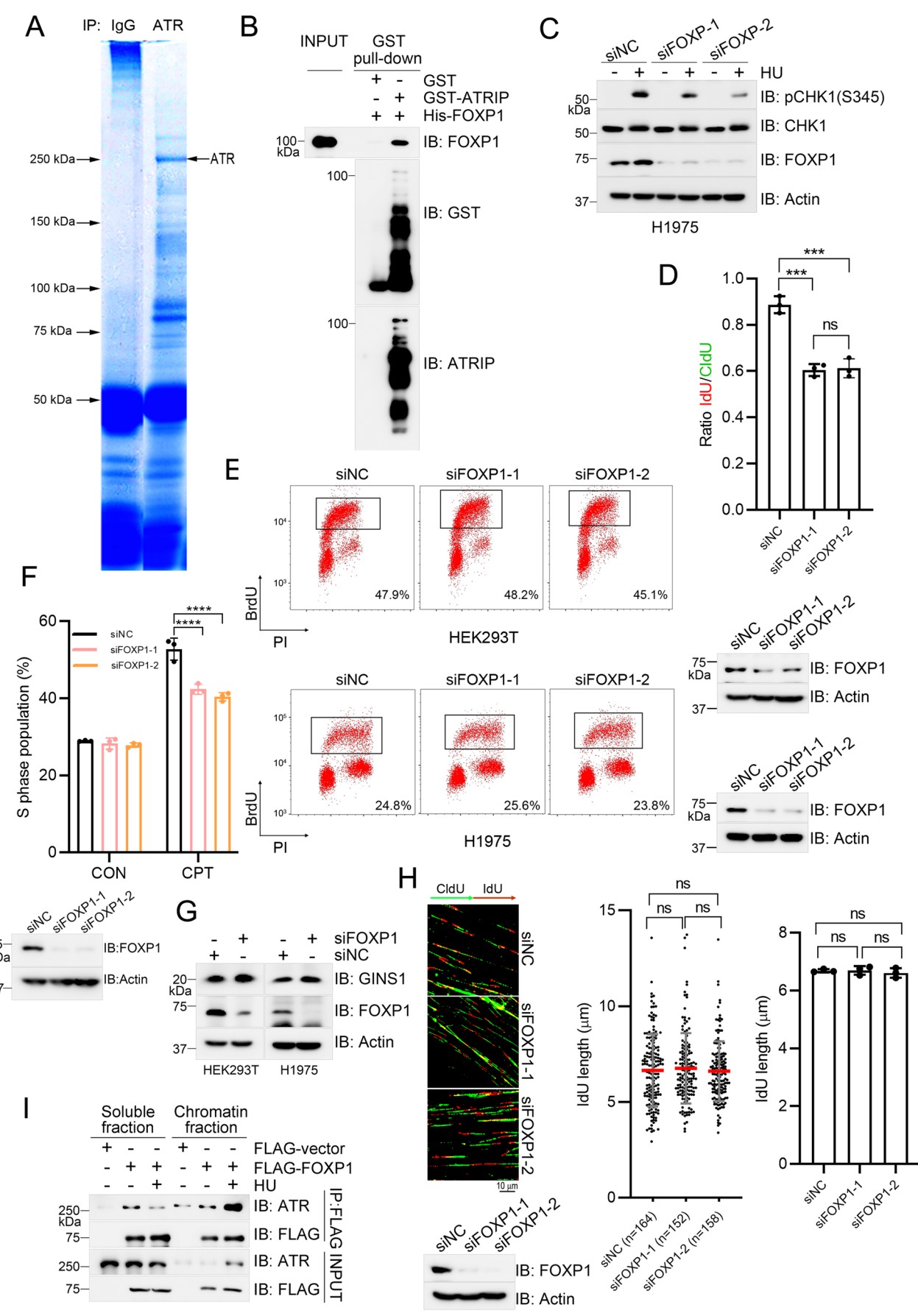

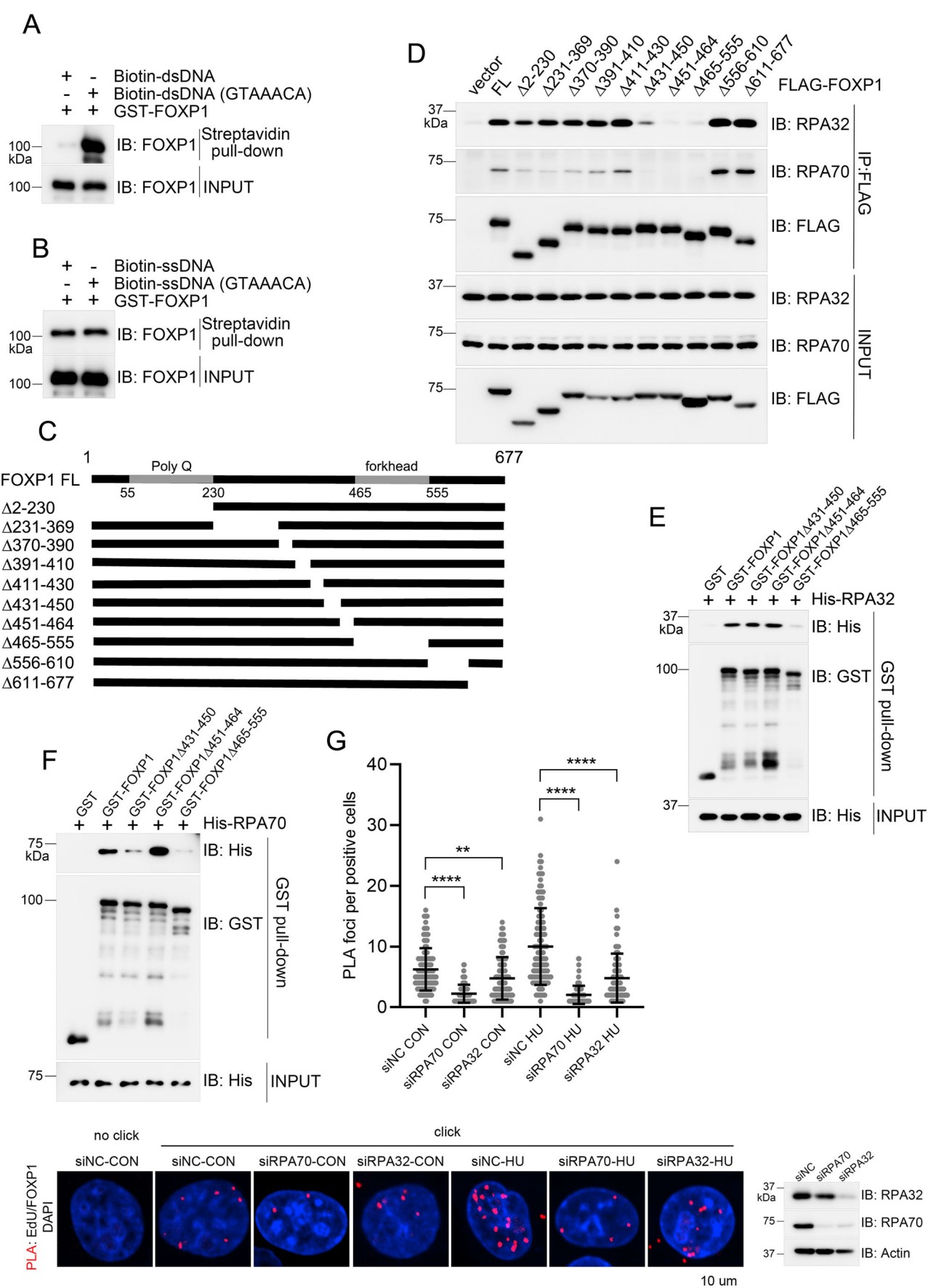

◀ **Figure EV2. Interaction mapping between FOXP1 and RPA.**

(A) Biotin-labeled random dsDNA or dsDNA containing GTAAACA consensus motif was conjugated on streptavidin magnetic beads and incubated with GST-tagged FOXP1 purified from *E. coli*. Streptavidin-bound FOXP1 was detected via immunoblotting with the indicated antibodies. (B) Biotin-labeled random ssDNA or ssDNA containing GTAAACA was used to test FOXP1 binding as described in (A). (C) Schematic of full-length FOXP1 and its deletion mutants. (D) HEK293T cells transfected with FLAG-FOXP1 or its deletion mutants were subjected to immunoprecipitation using a FLAG-specific antibody. FOXP1 bound RPA proteins were detected via immunoblotting. (E, F) His-tagged RPA32 (E) or RPA70 (F) were incubated with GST-tagged FOXP1 (or its deletion mutants) followed by GST pull-down assay. Proteins bound onto Glutathione beads were detected via immunoblotting using the indicated antibodies. (G) Proximity ligation assay experiments using FOXP1 and biotin-specific antibodies in H1975 cells transfected with a negative control siRNA or siRNAs targeting RPA32 or RPA70. Upper panel: quantification of the number of PLA foci per foci-positive cells (cell number: siNC CON, $n = 103$; siRPA70-CON, $n = 102$; siRPA32-CON, $n = 102$; siNC HU, $n = 110$; siRPA70-CON, $n = 106$; siRPA32-CON, $n = 106$), mean ± SD is shown. $**P < 0.01$, $****P < 0.0001$, $P$ values were calculated by one-way ANOVA, followed by Kruskal–Wallis test. $P$ value: siNC CON vs siRPA70-CON, 1.39e-018; siNC CON vs siRPA32-CON, 0.0039; siNC HU vs siRPA70 HU, 5.80e-038; siNC HU vs siRPA32 HU, 1.41e-010. Lower panel: representative images of PLA foci and immunoblotting of RPA expression. Source data are available online for this figure.

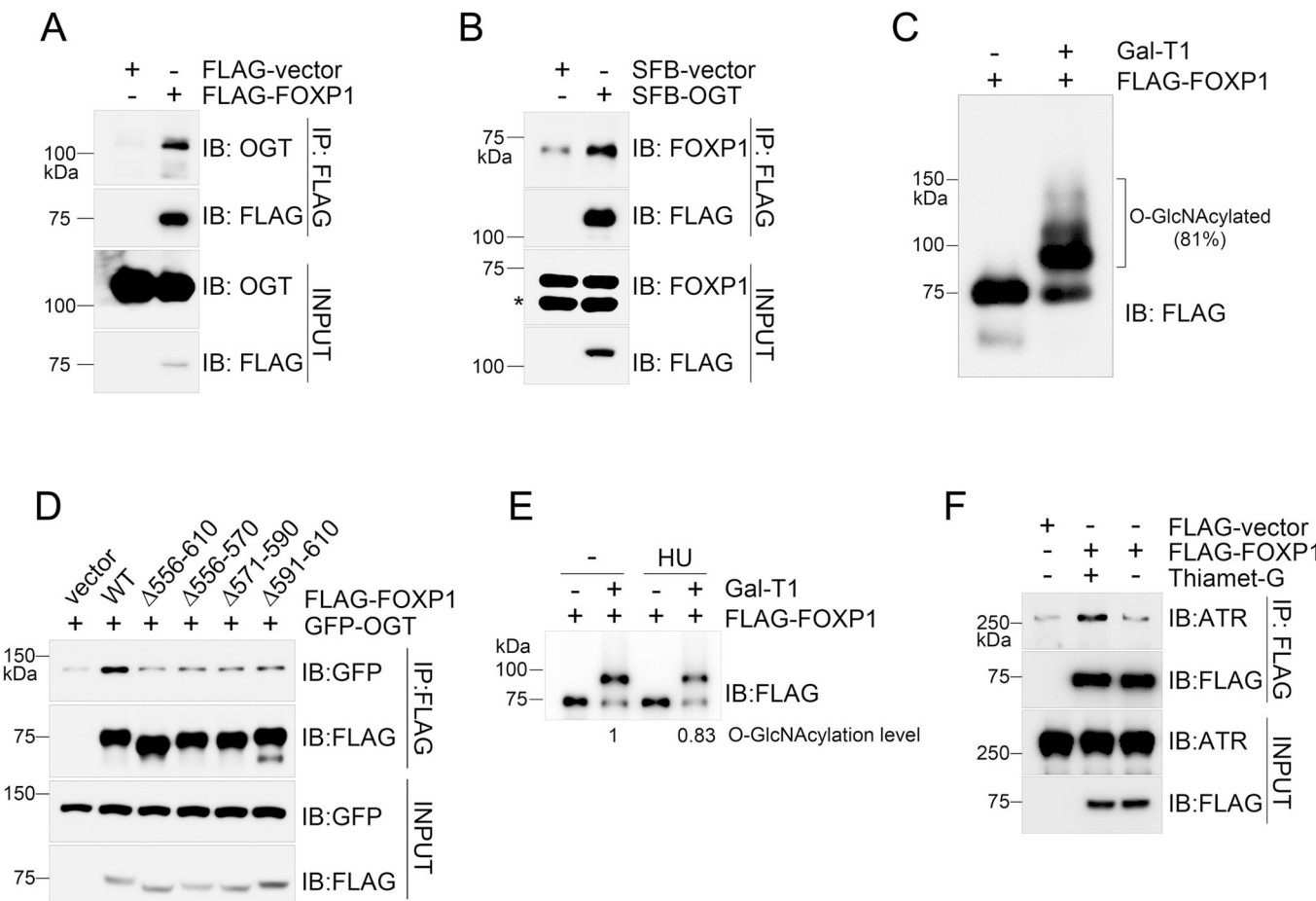

**Figure EV3.  O-GlcNAcylation of FOXP1 represses its interaction with ATR.**

(A, B) HEK293T cells transfected with FLAG-tagged FOXP1 (A) or SFB-tagged OGT (B) were subjected to immunoprecipitation using a FLAG-specific antibody. Proteins in the immunoprecipitates were detected via immunoblotting. * a shorter isoform of FOXP1. (C) O-GlcNAcylation stoichiometry of FLAG-tagged FOXP1 was analyzed and examined via immunoblotting with FLAG-specific antibody. (D) HEK293T cells transfected with GFP-tagged OGT and full-length FLAG-FOXP1 or its deletion mutants were subjected to immunoprecipitation and immunoblotting using the indicated antibodies. (E) O-GlcNAcylation stoichiometry of FLAG-tagged FOXP1 was analyzed and examined via immunoblotting with FLAG-specific antibody, with or without incubation with 2 mM HU for 1 h before cells were harvested. (F) H1975 cells transfected with FLAG-tagged FOXP1 preincubated with Thiamet-G or not were subjected to immunoprecipitation and immunoblotting using the indicated antibodies. Source data are available online for this figure.

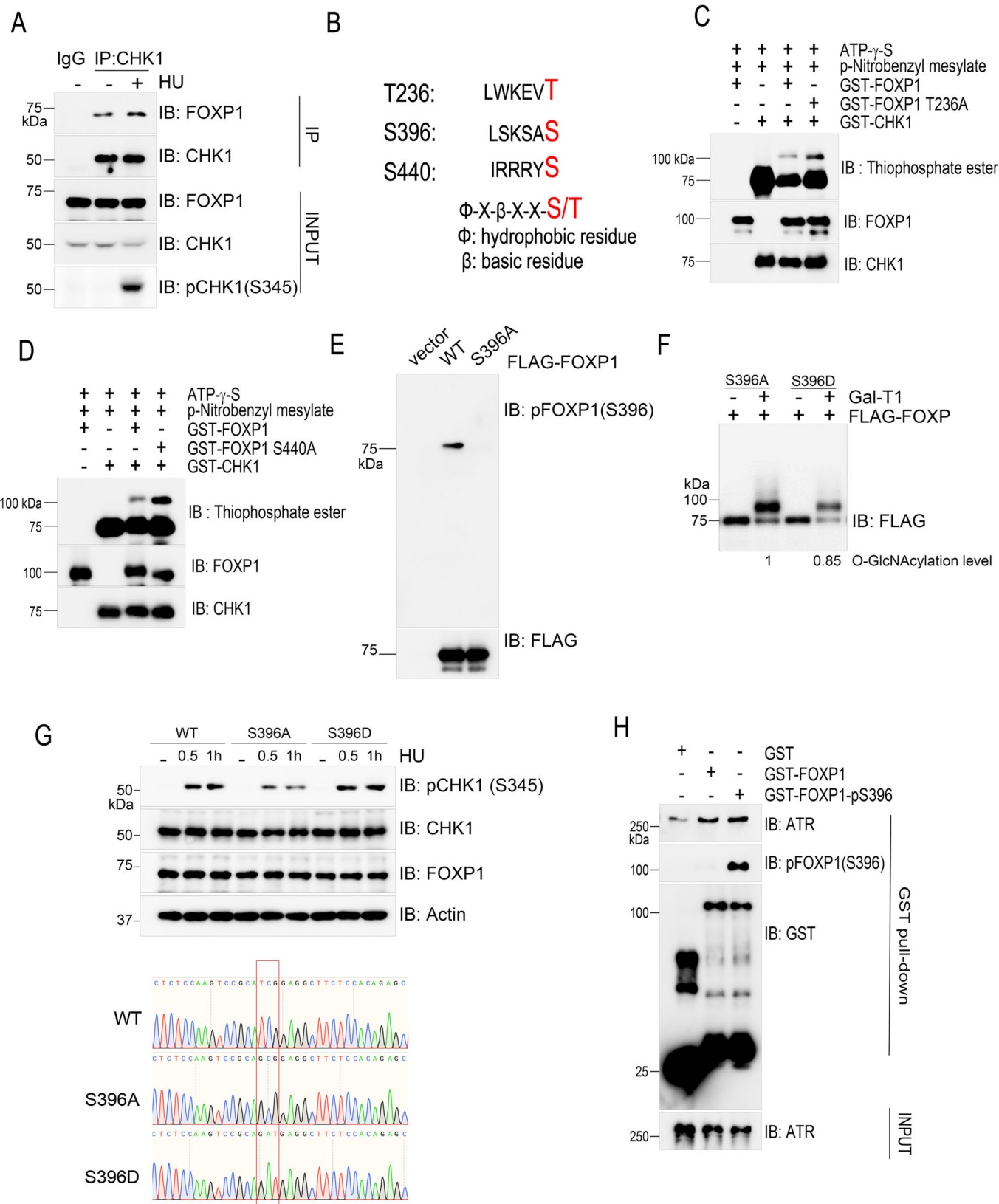

**Figure EV4.  CHK1-mediated FOXP1 phosphorylation at S396 promotes ATR activation.**

(**A**) HEK293T cells treated with 2 mM HU for 1 h or untreated were subjected to immunoprecipitation and immunoblotting using the indicated antibodies. (**B**) The potential phosphorylation motif recognized by CHK1 on FOXP1. (**C, D**) In vitro phosphorylation of FOXP1 and its T236A (**C**) or S440A mutant (**D**) by CHK1 were conducted using ATR-γ-S as the phosphor group donor. (**E**) HEK293T cells transfected with FLAG-tagged FOXP1 or S396A mutant were subjected to immunoprecipitation using a FLAG-specific antibody. The immunoprecipitates were examined via immunoblotting using a specific antibody to detect FOXP1 phosphorylation at S396. (**F**) O-GlcNAcylation stoichiometry of FLAG-tagged FOXP1 S396A or S396D were analyzed and examined via immunoblotting with FLAG-specific antibody. (**G**) Upper panel: Wild-type and S396A or S396D knock-in HEK293 cells were treated with HU for the indicated time before the whole cell lysates were harvested for immunoblotting with the indicated antibodies. Lower panel: genome sequence of wild-type and mutant HEK293 cell lines. (**H**) GST-tagged FOXP1 were subjected to in vitro kinase assay catalyzed by CHK1 or not, followed by incubation with HEK293T cell lysate before GST pulldown. The glutathione bead-bound signals were detected by immunoblotting. Source data are available online for this figure.

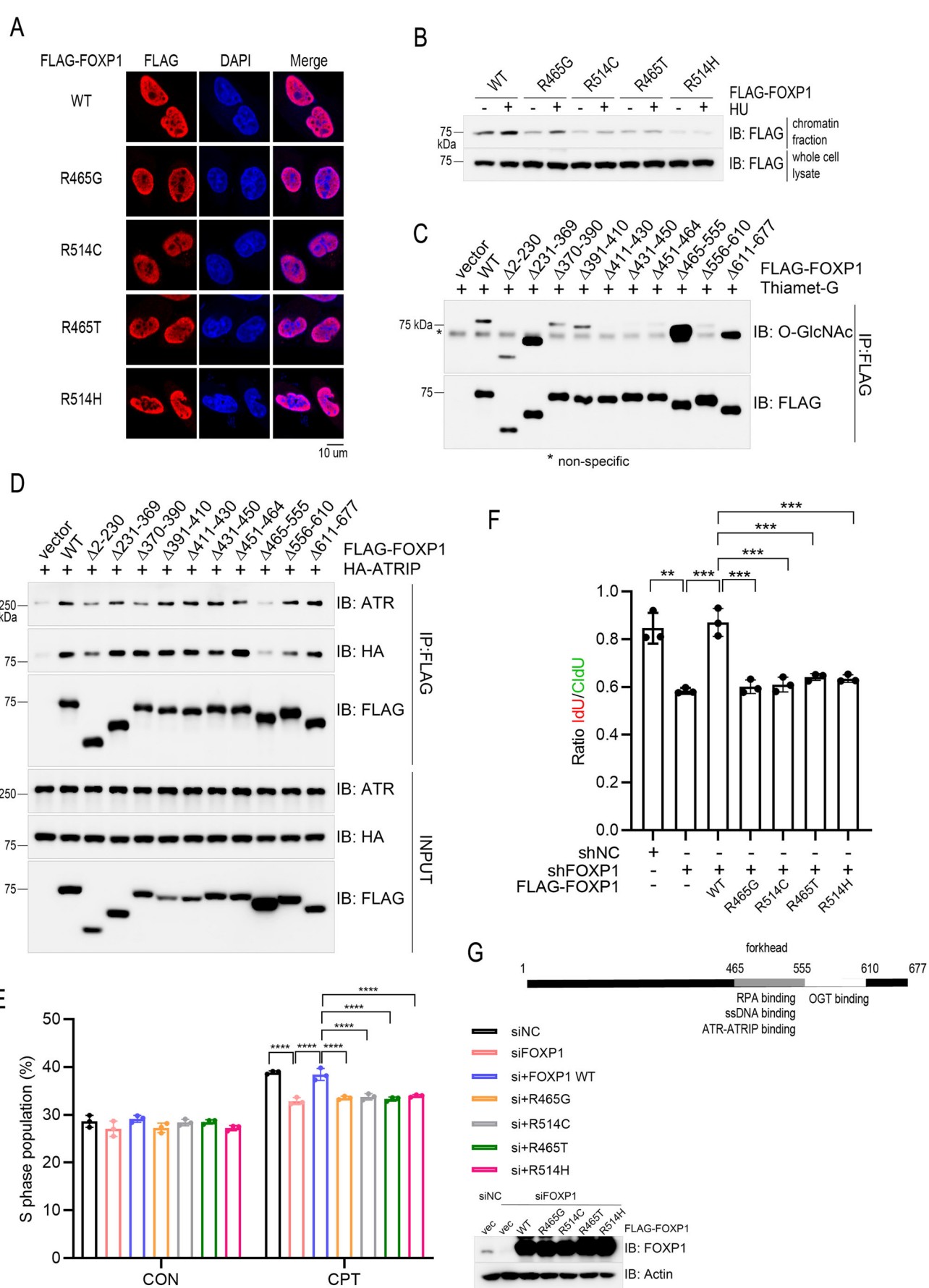

◀    **Figure EV5.    The FOXP1 forkhead domain is essential for its function during conditions of replication stress.**

(A) H1975 cells expressing FLAG-tagged FOXP1 or its mutants were subjected to immunostaining using FLAG-specific antibody, and the nuclei were detected with DAPI. (B) HEK293T cells transfected with FLAG-tagged FOXP1 or its mutants were treated with 2 mM HU for 1 h before chromatin fractionation. (C) HEK293T cells transfected with FLAG-tagged FOXP1 or its deletion mutants were preincubated with Thiamet-G before immunoprecipitation and then immunoblotting using the indicated antibodies. *, non-specific signal. (D) HEK293T cells transfected with FLAG-tagged FOXP1 or its deletion mutants and HA-tagged ATRIP were subjected to immunoprecipitation and then immunoblotting using the indicated antibodies. (E) FOXP1 knockdown HEK293 cells transfected with wild-type FLAG-FOXP1 or its pathogenic mutants were incubated with 100 nM CPT for 8 h or left untreated before harvested for PI staining and flow cytometric analysis. The percentage of S phase population was analyzed, mean ± SD (biological replicates, $n = 3$) is shown. ****$P < 0.0001$, $P$ values were calculated by two-way ANOVA, followed by Sidak's test. $P$ value: siNC vs siFOXP1, 2.75e-008; siFOXP1 vs siFOXP1 + FOXP1 WT, 1.28e-007; siFOXP1 + FOXP1 WT vs siFOXP1 + R465G, 1.50e-006; siFOXP1 + FOXP1 WT vs siFOXP1 + R514C, 3.66e-006; siFOXP1 + FOXP1 WT vs siFOXP1 + R465T, 6.25e-007; siFOXP1 + FOXP1 WT vs siFOXP1 + R514H, 7.92e-006. (F) Statistical analysis of the IdU/CldU ratio mean in Fig. 5F, mean ± SD (biological replicates, $n = 3$) is shown. **$P < 0.01$, ***$P < 0.001$, $P$ values were calculated by one-way ANOVA, followed by Dunnett's test. $P$ value: shNC vs shFOXP1, 0.0013, shFOXP1 vs shFOXP1+WT, 0.0008; shFOXP1+WT vs shFOXP1 + R465G, 0.0004; shFOXP1+WT vs shFOXP1 + R514C, 0.0005; shFOXP1+WT vs shFOXP1 + R465T, 0.0004; shFOXP1+WT vs shFOXP1 + R514H, 0.0004. (G) Functions of FOXP1 domains identified in this study. Source data are available online for this figure.

