## [Peer Review File · The EMBO Journal]

FOXP1 phosphorylation/O-GlcNAcylation in regulating ATR activation in response to replication stress

Xuefei Zhu, Congwen Gao, Bin Peng, Jingwei Xue, Donghui Xia, Liu Yang, Jiexiang Zhang, Xinrui Gao, Yilin Hu, Shixian Lin, Peng Gong, and Xingzhi Xu

Corresponding author(s): Xingzhi Xu (Xingzhi.xu@szu.edu.cn) , Peng Gong (doctorgongpeng@szu.edu.cn), Xuefei Zhu (zhuxuefei@szu.edu.cn)

Review Timeline:

Submission Date:	18th Apr 24
Editorial Decision:	15th May 24
Revision Received:	12th Oct 24
Editorial Decision:	5th Nov 24
Revision Received:	13th Nov 24
Accepted:	18th Nov 24

Editor: Hartmut Vodermaier

Transaction Report:

Dr. Xingzhi (Xavier) Xu
School of Medicine, Shenzhen University
Cell Biology
1066 Xueyuan Blvd, Rm. A6-901
Shenzhen University Lihu Campus
Shenzhen, Guangdong 518055
China

15th May 2024

Re: EMBOJ-2024-117634
FOXP1 phosphorylation/O-GlcNAcylation in regulating ATR activation in response to replication stress

Dear Xavier,

Thank you for submitting your manuscript on FOXP1 phosphorylation and O-GlcNAcylation in the replication stress response to The EMBO Journal. I sent it to three expert referees, who have now returned the reports that are copied below. As you will see, the referees consider your findings potentially interesting and the analyses for the most part well done. Nevertheless, they raise a number of important concerns with the conclusiveness of some of the data, which would in our view have to be decisively clarified before publication.

Should you be able to adequately address the referees' concerns, we would be happy to pursue a revised manuscript further for EMBO Journal publication. However, I should emphasize that we only allow a single round of major revision, and this makes it important to fully and carefully respond to each referee point at the time of resubmission. I would therefore encourage you to contact us with a revision plan, preliminary point-by-point response, or any other questions you may have in this regard already during the early stages of your revision work, in order to clarify if and how key issues raised in the reports may be solved. We would also be open to extension of the regular three-months revision period if needed; our 'scooping protection' (meaning that competing work appearing elsewhere in the meantime will not affect our considerations of your study) would of course remain valid also throughout such an extension.

Further information on preparing, formatting and uploading a revised manuscript can be found below and in our Guide to Authors. Thank you again for the opportunity to consider this work for The EMBO Journal, and I look forward to your revision.

With kind regards,

Hartmut

3) Revised manuscript text (including main tables, and figure legends for main and EV figures) has to be submitted as editable

text file (e.g., .docx format). We encourage highlighting of changes (e.g., via text color) for the referees' reference.

4) Each main and each Expanded View (EV) figure should be uploaded as individual production-quality files (preferably in .eps, .tif, .jpg formats). For suggestions on figure preparation/layout, please refer to our Figure Preparation Guidelines:

8) Please note that supplementary information at EMBO Press has been superseded by the 'Expanded View' for inclusion of additional figures, tables, movies or datasets; with up to five EV Figures being typeset and directly accessible in the HTML version of the article. For details and guidance, please refer to:

embopress.org/page/journal/14602075/authorguide#expandedview

9) Digital image enhancement is acceptable practice, as long as it accurately represents the original data and conforms to community standards. If a figure has been subjected to significant electronic manipulation, this must be clearly noted in the figure legend and/or the 'Materials and Methods' section. The editors reserve the right to request original versions of figures and the original images that were used to assemble the figure. Finally, we generally encourage uploading of numerical as well as gel/blot image source data; for details see: embopress.org/page/journal/14602075/authorguide#sourcedata

At EMBO Press, we ask authors to provide source data for the main manuscript figures. Our source data coordinator will contact you to discuss which figure panels we would need source data for and will also provide you with helpful tips on how to upload and organize the files.

Further information is available in our Guide For Authors:

In the interest of ensuring the conceptual advance provided by the work, we recommend submitting a revision within 3 months (13th Aug 2024). Please discuss the revision progress ahead of this time with the editor if you require more time to complete the revisions. Use the link below to submit your revision:

Link Not Available

Referee #1:

The manuscript "FOXP1 phosphorylation/O-GlcNAcylation in regulating ATR activation in response to replication stress" by Zhu et al., describes a novel mechanism of ATR activation in response to replication stress, mediated by FOXP1. The authors additionally identified O-GlcNAcylation-dependent and CHK1-phosphorylation-dependent regulation of this mechanism and characterized several disease-associated mutants of FOXP1. The story is novel, interesting, and would be a valuable addition to our understanding of ATR regulation in response to replication stress. However, several concerns should be addressed before publication.

1. Most of the evidence comes from biochemical assays that clearly demonstrate protein-protein interactions and their regulation by various mutations. The main weakness of this manuscript, in my opinion, is insufficient cell biology data to support the role of the identified interactions in response to replication stress and at the replication forks. To my knowledge, there are dozens of iPOND datasets available, can FOXP1 recruitment to stalled forks be observed at least in the existing datasets? The provided PLA (2G) lacks negative controls and is difficult to interpret (more below). CHK1-dependent phosphorylation, so clear in the in vitro assays, is barely induced by replication stress in cells (fig. 4G).

2. The major function of ATR is in S-phase checkpoint and cell cycle arrest. The inability of cells to activate ATR should result in inability to stop the cell cycle in response to DNA damage. It could also result in double strand DNA breaks and ATR activation. This is a better readout than fork stability, as fork stability is regulated by dozens of different factors that could potentially be disrupted by FOXP1 deficiency. Inability to activate cell cycle arrest should be demonstrated with FOXP1 siRNAs and with

FOXP1 mutants.

3. Does this pathway work as an alternative to TOPBP1 and ETAA1-mediated pathways, or do they work in parallel? These two well described pathways seem to not require any additional help to recruit ATR to RPA-covered DNA and have been reconstituted *in vitro*. Does FOXP1 have an ATR-activating domain? Discussion does not provide the context for this novel ATR-activation pathway. It should be compared to the known pathways and a model how all these pathways co-exist should be proposed, and, ideally, tested.

4. The effects of HU are very mild throughout the study (effect on interactions, effect on FOXP1 phosphorylation, effect on chromatin loading, etc.), and barely noticeable compared to the ATR-induced phosphorylations, CHK1 p345 for example, that are very robust. The authors should consider if FOXP1 participates in the base-line ATR activation in the absence of replication stress, that controls the number of active origins. If base-line ATR activation is disrupted, the authors should observe increased origin firing in FOXP1-disrupted cells.

5. To assess the role of O-GlcNAcylation in ATR activity regulation, it would be nice to see the readout for actual ATR activity when O-GlcNAcylation is disrupted by siRNA against OGT. Does it cause increased ATR activation (CHK1 p345), even in the absence of HU (as ATR binding to chromatin is increased without HU (fig. 3H)), and if so, could OGT depletion be causing replication stress independently of HU? This could be tested by detecting ssDNA in cells using native BrdU IF.

6. Fig. 1 D-F. siFOXP1-1 is not as efficient for FOXP1 depletion as siFOXP1-2, but for some reason shows more significant effect on fork stability. How do authors reconcile these observations? Have fiber experiments been repeated three times (in this case, error bars should be added to the graph)?

7. Fig. 2G. The authors should show single-antibody controls for PLA - this method is well known for a difficult to eliminate background. The effect of HU again is not very robust here, but also just about 10 co-localization events per cell seems a little low for HU-induced replication stress where every fork is a stalled fork. Some stronger evidence is needed to conclude that FOXP1 is recruited to stalled forks, ideally an iPOND.

8. Fig 5E shows two FOXP1 western blots, labeled identically, one of which was cut in a strange way. I suspect that it could be long exposure and short exposure, but it has to be clarified and the cut area should be shown.

9. Some inconsistencies with CHK1 inhibitors need more explanations: classic UCN01 does not fully inhibit the interaction of FOXP1 with ATR, while rabusertib suppresses ATR-dependent CHK1 phosphorylation on S345 (Fig. 4b), indicating that it may actually inhibit ATR. Similarly, on fig 4G FOXP1 phosphorylation (4g) is fully suppressed by rabusertib, but UCN01 barely shows any effect. An antibody against CHK1 pS296 should be used as a readout of CHK1 activity for a positive control of the used compounds.

10. The authors claim that the FOXP1 mutations negatively affect ATR activation, but ATR activation was not tested with the mutants at all. The authors can only make conclusions about their effects on interactions and on fork stability. Not even CHK1 phosphorylation is shown.

Referee #2:

In this manuscript, Zhu. et. al., found that:

1, FOXP1 is recruited at stalled replication fork and regulates ATR activation.

2, FOXP1 is phosphorylated by CHK1 and O-GlcNacetylated by OGT. Crosstalk between these 2 PTM is essential for ATR activation during replication stress.

While many of the findings are interesting, more details are needed to convince the audience.

1, Which modification (phosphorylation or O-GlcNacetylation) dictates ATR regulation by FOXP1?

It seems that FOXP1 O-GlcNacetylation is most pivotal. Is phosphorylation merely a way to regulate O-GlcNacetylation? What does phosphorylation do to FOXP1 when it cannot be O-GlcNacetylated (to understand this, O-GlcNacetylation site should be identified)?

2, How does FOXP1 activate ATR? Does RPA regulate FOXP1's recruitment to stalled replication forks (Does phosphorylation or O-GlcNacetylation play a role in it)? If FOXP1 participate the RPA-ATR/ATRIP activation axis, what is the role of the known ATR activator TOPBP1 or ETAA1 in this? Can FOXP1 directly activate ATR/ATRIP (or with RPA) *in vitro*?

3, If O-GlcNacetylation decreasing after HU is the main reason for the subsequent FOXP1 recruitment, ATR binding and ATR activation, one must assume that FOXP1 O-GlcNacetylation is high enough to avoid abnormal ATR activation when without HU treatment. What is the stoichiometry of FOXP1 O-GlcNacetylation before and after HU treatment?

4, How is ATR activation in OGT depleted cells with or without HU? What if depleted with FOXP1 in OGT depleted cells?

5, Since FOXP1 also interact with ATR or RPA before HU, does FOXP1 affect ATR phosphorylation without HU treatment? In Fig. 1D, loss of FOXP1 compromised ATR phosphorylation without HU (ATR seems not be activated upon HU in this panel), but in supplementary Fig. 1C, no effect. If FOXP1 does not affect ATR phosphorylation without HU, but depletion of FOXP1 should compromised FOXP1-ATR/RPA binding, then ATR phosphorylation should still be diminished further in FOXP1 KD cells, even without HU? What is the author's explanation here.

6, Because FOXP1 is a transcription factor, does FOXP1 regulate ATR activation and protect stalled replication fork globally or at any preferred DNA regions?

Referee #3:

The authors show that FOXP1 mediates ATR-Chk1 signaling through a mechanism that is controlled by O-GlcNAcylation or phosphorylation of a single serine residue. ATR-Chk1-phosphorylation on FOXP1 serine-396 potentiates the association of RPA-ATR-ATRIP and Chk1 activation. O-GlcNAcylation on FOXP1 serine-396 reduces the association of RPA-ATR-ATRIP. The paper is well written, and the figures are well presented.

Major concern:

Figure 2G shows very little difference between baseline and HU-induced EdU-Biotin FOXP1 PLA foci. Can the authors speak to the stoichiometry of FOXP1 phosphorylation vs O-GlcNAcylation vs unmodified serine-396?

Is there direct evidence that endogenous FOXP1 is subject to O-GlcNAcylation in vivo? I see ectopic expression of tagged proteins in 293T cells.

Minor concern:

Inconsistent results between Fig 1D and Fig 1H where in the former, FOXP1-2 knockdown has the greatest impact on FOXP1 function, while in the latter, FOXP1-1 has the greatest impact on FOXP1 function. Dose and time titrations are required to speak to activity.

The paper would be significantly improved by knockin mutations of FOXP1 alanine-396 or glutamic acid-396.

Point-to-point responses:**Referee #1:**

The manuscript "FOXPI phosphorylation/O-GlcNAcylation in regulating ATR activation in response to replication stress" by Zhu et al., describes a novel mechanism of ATR activation in response to replication stress, mediated by FOXPI. The authors additionally identified O-GlcNAcylation-dependent and CHK1-phosphorylation-dependent regulation of this mechanism and characterized several disease-associated mutants of FOXPI. The story is novel, interesting, and would be a valuable addition to our understanding of ATR regulation in response to replication stress. However, several concerns should be addressed before publication.

1. Most of the evidence comes from biochemical assays that clearly demonstrate protein-protein interactions and their regulation by various mutations. The main weakness of this manuscript, in my opinion, is insufficient cell biology data to support the role of the identified interactions in response to replication stress and at the replication forks. To my knowledge, there are dozens of iPOND datasets available, can FOXPI recruitment to stalled forks be observed at least in the existing datasets? The provided PLA (2G) lacks negative controls and is difficult to interpret (more below). CHK1-dependent phosphorylation, so clear in the in vitro assays, is barely induced by replication stress in cells (fig. 4G).

Response: Thank you for highlighting this. We have identified a couple of iPOND datasets that show FOXPI present on replication forks or stalled replication forks [1] [2, 3]. Additionally, we confirmed the recruitment of FOXPI to stalled replication forks using PLA assays (Fig. 2G) and iPOND assays (Fig. 2H in the revised version), as mentioned in our response to Q7. The concerns regarding Fig. 2G and Fig. 4G have been addressed in our responses to Q7 and Q9, respectively.

2. The major function of ATR is in S-phase checkpoint and cell cycle arrest. The inability of cells to activate ATR should result in inability to stop the cell cycle in

response to DNA damage. It could also result in double strand DNA breaks and ATR activation. This is a better readout than fork stability, as fork stability is regulated by dozens of different factors that could potentially be disrupted by FOXP1 deficiency. Inability to activate cell cycle arrest should be demonstrated with FOXP1 siRNAs and with FOXP1 mutants.

Response: Thank you for your thoughtful suggestions. We used a low dose of camptothecin (CPT) to induce replication stress for cell cycle arrest analysis. FOXP1 knockdown cells also showed reduced ATR activation under CPT-induced replication stress (Graphic 1). We collected negative control or FOXP1 knockdown HEK293 cells for flow cytometry analysis after incubation with 100 nM CPT for 8 hours. The results indicated that FOXP1-deficient cells were less efficient at arresting in the S phase. Additionally, HEK293 cells expressing FOXP1 pathogenic mutants showed impaired S phase arrest compared to those expressing wildtype FOXP1. These data are included as Fig. S1F and S5E, respectively, in the revised manuscript.

Graphic 1

Figure S1F

Figure S5E

3. Does this pathway work as an alternative to TOPBP1 and ETAA1-mediated pathways, or do they work in parallel? These two well described pathways seem to not require any additional help to recruit ATR to RPA-covered DNA and have been

reconstituted in vitro. Does FOXP1 have an ATR-activating domain? Discussion does not provide the context for this novel ATR-activation pathway. It should be compared to the known pathways and a model how all these pathways co-exist should be proposed, and, ideally, tested.

Response: Thank you for your thoughtful suggestions. In our working model, FOXP1 acts as a scaffold protein to facilitate the recruitment of ATR, which is subsequently activated by TopBP1 and ETAA1. Although FOXP1 promotes the chromatin loading of ATR, the loading of TOPBP1 and ETAA1 was not affected in FOXP1-depleted cells. This data is shown in Fig. 1H of the revised manuscript. Additionally, FOXP1 does not contain a clear ATR-activating domain like those found in TopBP1 and ETAA1, as previously reported [4]. We also performed an *in vitro* kinase assay using FLAG-tagged ATR/ATRIP expressed in HEK293T cells, with His-tagged CHK1 as the substrate, and found that the pCHK1S345 level did not increase in the presence of GST-tagged FOXP1 (Graphic 2). We propose that FOXP1 serves as a scaffold protein for optimized ATR recruitment, a prerequisite for its activation, but does not directly regulate the ATR allosteric activation step by TopBP1 and ETAA1. This is discussed in the revised manuscript.

Figure 1H

Graphic 2

4. The effects of HU are very mild throughout the study (effect on interactions, effect on FOXP1 phosphorylation, effect on chromatin loading, etc.), and barely noticeable compared to the ATR-induced phosphorylations, CHK1 p345 for example, that are

very robust. The authors should consider if *FOXP1* participates in the base-line ATR activation in the absence of replication stress, that controls the number of active origins. If base-line ATR activation is disrupted, the authors should observe increased origin firing in *FOXP1*-disrupted cells.

Response: Thank you for your thoughtful suggestions. We tested baseline ATR activation by examining pCHK1S345 levels in unperturbed negative control and *FOXP1*-depleted cells and observed similar pCHK1S345 levels in both cell lines (Graphic 3). Additionally, we assessed DNA replication using a DNA fiber assay and found that the inter-origin distance in both negative control and *FOXP1*-depleted cells was comparable (Graphic 4).

Graphic 3

Graphic 4

5. To assess the role of O-GlcNAcylation in ATR activity regulation, it would be nice to see the readout for actual ATR activity when O-GlcNAcylation is disrupted by siRNA against OGT. Does it cause increased ATR activation (*CHK1* p345), even in the absence of HU (as ATR binding to chromatin is increased without HU (fig. 3H)), and if so, could OGT depletion be causing replication stress independently of HU? This could be tested by detecting ssDNA in cells using native BrdU IF.

Response: We appreciate your suggestions. When O-GlcNAcylation was disrupted using siRNA against OGT, pCHK1S345 levels decreased both in the unperturbed state and after HU treatment (Graphic 5). Since OGT is the only known O-GlcNAc transferase in mammalian cells, its depletion disrupts O-GlcNAcylation homeostasis

and may affect the function of numerous substrates, aside from FOXP1, which could directly or indirectly regulate the replication stress response. Figure 3H illustrates the interaction between ATR and FOXP1, but not ATR binding to chromatin. Additionally, we tested ssDNA generation using native BrdU immunofluorescence and found that OGT depletion did not affect ssDNA generation either before or after HU treatment (Graphic 6).

Graphic 5

Figure 3H

Graphic 6

6. Fig.1 D-F. *siFOXP1-1* is not as efficient for *FOXP1* depletion as *siFOXP1-2*, but for some reason shows more significant effect on fork stability. How do authors reconcile these observations? Have fiber experiments been repeated three times (in this case, error bars should be added to the graph)?

Response: Thank you for your question. Experiment in Fig. 1D is independent of that in Figs. 1E-F. We found that the knockdown efficiency between *siFOXP1* oligos could be variable from experiment to experiment. We repeated this experiment in one more time and achieved similar knockdown efficiency between *siFOXP1-1* and *siFOXP1-2* along with similar degrees of reduction of CHK1 activation (Fig. 1D).

In Fig. 1E, the knockdown efficiency of FOXP1 using siFOXP1-2 was slightly better than with siFOXP1-1; however, statistical analysis showed no significant difference in fork stability between these two groups in Fig. 1F. Data from two repeated experiments has been integrated into the revision (Figs. S1D, S1H, and S5F). The mean \pm SD values have been added to the scatter plots in Figs. 1F, S1H, and 5G.

7. Fig. 2G. The authors should show single-antibody controls for PLA - this method is well known for a difficult to eliminate background. The effect of HU again is not very robust here, but also just about 10 co-localization events per cell seems a little low for HU-induced replication stress where every fork is a stalled fork. Some stronger evidence is needed to conclude that FOXP1 is recruited to stalled forks, ideally an *iPOND*.

Response: We appreciate your suggestions. We repeated the PLA assay in Fig. 2G, included single-antibody controls, and extended the HU treatment. Additionally, we

performed the iPOND experiment and confirmed the recruitment of FOXP1 to stalled replication forks. This data is included in Fig. 2H of the revised manuscript.

Figure 2G

Figure 2H

8. Fig 5E shows two FOXP1 western blots, labeled identically, one of which was cut in a strange way. I suspect that it could be long exposure and short exposure, but it has to be clarified and the cut area should be shown.

Response: Thank you for pointing this out. We apologize for the incomplete presentation of the short FOXP1 western blot, which only included the first four lanes. A longer exposure of all samples should have been provided. We have since reperformed this experiment, as detailed in our response to Q10.

9. Some inconsistencies with CHK1 inhibitors need more explanations: classic UCN01 does not fully inhibit the interaction of FOXP1 with ATR, while rabusertib suppresses ATR-dependent CHK1 phosphorylation on S345 (Fig. 4b), indicating that it may actually inhibit ATR. Similarly, on fig 4G FOXP1 phosphorylation (4g) is fully suppressed by rabusertib, but UCN01 barely shows any effect. An antibody against CHK1 pS296 should be used as a readout of CHK1 activity for a positive control of the used compounds.

Response: Thank you for bringing this to our attention. We repeated the experiments in Fig. 4B and Fig. 4G with optimized working concentrations of Rabusertib and

UCN01 in the revised manuscript. The concentration of Rabusertib was reduced from 10 μ M to 5 μ M, while that of UCN01 was increased from 10 nM to 50 nM. We also examined pCHK1S296 signals, which confirmed effective inhibition of CHK1 activity by these two inhibitors.

Figure 4B

Figure 4G

10. The authors claim that the FOXP1 mutations negatively affect ATR activation, but ATR activation was not tested with the mutants at all. The authors can only make conclusions about their effects on interactions and on fork stability. Not even CHK1 phosphorylation is shown.

Response: We appreciate your thoughtful suggestions. In the revised manuscript, we tested pCHK1S345 levels in cells expressing FOXP1 mutations. The results showed that both the pathogenic mutants and the S396A mutant are deficient in promoting ATR activation compared to wildtype FOXP1. These data are included in the updated Fig. 5E and Fig. S4F (now Fig. S4G in the revision).

Figure 5E

Figure S4G

Referee #2:

In this manuscript, Zhu. et. al., found that:

- 1, FOXP1 is recruited at stalled replication fork and regulates ATR activation.*
 - 2, FOXP1 is phosphorylated by CHK1 and O-GlcNacylated by OGT. Crosstalk between these 2 PTM is essential for ATR activation during replication stress.*
- While many of the findings are interesting, more details are needed to convince the audience.*

1, Which modification (phosphorylation or O-GlcNacylation) dictates ATR regulation by FOXP1?

It seems that FOXP1 O-GlcNacylation is most pivotal. Is phosphorylation merely a way to regulate O-GlcNacylation? What does phosphorylation do to FOXP1 when it cannot be O-GlcNacylated (to understand this, O-GlcNacylation site should be identified)?

Response: Thank you for the insightful suggestions. We initially attempted to identify the O-GlcNAcylation site(s) on FOXP1 using mass spectrometry, which predicted S448 as a strong candidate (Graphic 7), with T415 and T418 also being potential sites. However, we did not observe a notable decrease in FOXP1 O-GlcNAcylation levels when testing individual residue mutants or the T415A/T418A/S448A mutant (Graphic 8 and Graphic 9). We also searched the public database (<https://www.oglcnac.mcw.edu/>), which predicts multiple potential O-GlcNAcylation sites on FOXP1, including S329, T408, T411, T415, T418, S422, T427, S440, S448, and S449. Stoichiometry analysis of FOXP1 (Fig. S3C in the revision) in our response to Q3 also indicated that FOXP1 is modified by O-GlcNAcylation at multiple sites.

Our examination of FOXP1 deletion mutants for O-GlcNAcylation revealed that deletion of AA 556-610 significantly reduced O-GlcNAcylation, however, AA 556-610 mediates the interaction between FOXP1 and OGT. Meanwhile, deletion of AA 370-464 also significantly reduced O-GlcNAcylation (Fig. S5C). This region is enriched with more than twenty serine and threonine residues, all of which are

potential O-GlcNAcylation sites. As a result, pinpointing the specific serine or threonine residues responsible for FOXP1 O-GlcNAcylation without affecting other functions of FOXP1 is quite challenging.

To explore whether FOXP1 phosphorylation influences its binding to ATR when O-GlcNAcylation is absent, we purified GST-tagged FOXP1 without O-GlcNAcylation from *E.coli* and performed a CHK1-mediated FOXP1 phosphorylation assay. We then conducted a GST pulldown experiment using unmodified and phosphorylated FOXP1. The results showed similar ATR levels interacting with S396-phosphorylated and unmodified FOXP1 (Fig. S4H in the revised manuscript). In contrast, O-GlcNAcylated GST-tagged FOXP1 exhibited reduced binding to ATR compared to unmodified FOXP1 (Fig. 3I). These data collectively indicate that O-GlcNAcylation of FOXP1 directly inhibits its interaction with ATR, while phosphorylation indirectly promotes the interaction by repressing O-GlcNAcylation.

Graphic 7

Graphic 8

Graphic 9

Fig.5C

Fig.S4H

2, How does FOXP1 activate ATR? Does RPA regulate FOXP1's recruitment to stalled replication forks (Does phosphorylation or O-GlcNAcylation play a role in it)?

If FOXP1 participate the RPA-ATR/ATRIP activation axis, what is the role of the known ATR activator TOPBP1 or ETAA1 in this? Can FOXP1 directly activate ATR/ATRIP (or with RPA) in vitro?

Response: Thank you for the helpful suggestions. We found that FOXP1 loading onto stalled replication forks was reduced in RPA knockdown cells, indicating that FOXP1 recruitment is RPA-dependent. This data is included in Fig. S2G of the revised manuscript. Additionally, in vitro binding assays showed that bacterially-produced FOXP1 binds directly to both ssDNA and RPA (Figs. 2A-2E), suggesting that neither phosphorylation nor O-GlcNAcylation is required for its binding to RPA. In fact, O-GlcNAcylation and phosphorylation of FOXP1 did not affect its binding to RPA. Using unmodified or O-GlcNAcyated GST-tagged FOXP1 in GST pulldown assays, we detected that both forms exhibited similar affinity for RPA. Likewise, FOXP1 phospho-deficient and mimic mutants (S396A and S396D) also showed similar binding capacity to RPA (Graphics 10 and 11).

Figure S2G

Graphic 10

Graphic 11

In our working model, FOXP1 acts as a scaffold protein to facilitate the recruitment of ATR, which is subsequently activated by TopBP1 and ETAA1. Although FOXP1 promotes the chromatin loading of ATR, the loading of TOPBP1 and ETAA1 was unaffected in FOXP1-depleted cells, as shown in Fig. 1H of the revised manuscript. Furthermore, FOXP1 lacks an obvious ATR activation domain, such as those present

in TopBP1 and ETAA1, as previously reported [4].

We performed an in vitro kinase assay using FLAG-tagged ATR/ATRIP expressed in HEK293T cells, with His-tagged CHK1 as the substrate. While FOXP1 can directly bind to ATR/ATRIP (Fig. S1B), the pCHK1S345 level was not elevated in the presence of GST-tagged FOXP1 (Graphic 2). This suggests that FOXP1 functions as a scaffold protein for optimal ATR recruitment, a prerequisite for its activation, but does not directly regulate the ATR allosteric activation step mediated by TopBP1 and

Figure 1H

Graphic 2

ETAA1.

3, *If O-GlcNacylation decreasing after HU is the main reason for the subsequent FOXP1 recruitment, ATR binding and ATR activation, one must assume that FOXP1 O-GlcNacylation is high enough to avoid abnormal ATR activation when without HU treatment. What is the stoichiometry of FOXP1 O-GlcNacylation before and after HU treatment?*

Response: Thank you for the valuable suggestions. We examined FOXP1 O-GlcNacylation stoichiometry using the chemoenzymatic labeling method [5]. Immunoprecipitated FLAG-FOXP1 was labeled with GalNAz and conjugated with DBCO-PEG5K. Our results showed that about 80 percent of FOXP1 was modified with O-GlcNacylation, and HU treatment led to a reduction in FOXP1 O-GlcNacylation. These findings are included in Fig. S3C and Fig. S3E of the

revised manuscript.

Fig.S3C

Fig.S3E

4, How is ATR activation in OGT depleted cells with or without HU? What if depleted with FOXP1 in OGT depleted cells?

Response: Thank you for your questions. ATR activation was reduced in OGT-depleted cells both before and after HU treatment (Graphic 12). Additionally, FOXP1 depletion further inhibited ATR activation in OGT-depleted cells (Graphic 13).

Graphic 12

Graphic 13

5, Since FOXP1 also interact with ATR or RPA before HU, does FOXP1 affect ATR phosphorylation without HU treatment? In Fig. 1D, loss of FOXP1 compromised ATR phosphorylation without HU (ATR seems not be activated upon HU in this panel), but in supplementary Fig. 1C, no effect. If FOXP1 does not affect ATR phosphorylation without HU, but depletion of FOXP1 should compromised FOXP1-ATR/RPA binding, then ATR phosphorylation should still be diminished further in FOXP1 KD cells, even without HU? What is the author's explanation here.

Response: Thank you for pointing this out. Fig. 1C was performed in HEK293T cells, while Fig. S1C was conducted in H1975 cells. In Fig. 1D, the ATR phosphorylation signal remained similar before and after HU treatment, whereas ATR-mediated CHK1

phosphorylation at S345 (pCHK1S345) was notably increased.

During the 5-month revision, we repeated this experiment several times using three different batches of pATR(T1989) antibodies from two independent companies (Abcam, ab223258, lot#1003610-2 and 1003610-9; CST, 30632S, lot#D5K8W-3), but we still did not detect a clear increase in the pATR(T1989) signal after HU treatment. However, we repeatedly observed that both the pCHK1S345 signal (new version of Figs. 1D, 5E, and S4G) and ATR loading onto chromatin (new version of Fig. 1H) increased obviously. Therefore, in the revised manuscript, we use the pCHK1S345 signal to indicate ATR activation under replication stress, as well as for the analysis of FOXP1 mutants, following the suggestion from Referee #1 (Q10).

We also tested ATR activation in the unperturbed state and observed similar ATR activation levels with or without FOXP1 depletion. The low levels of endogenous replication stress in unperturbed conditions may limit FOXP1's function in this context (Graphic 3).

Figure 1D

Graphic 3

6, Because FOXP1 is a transcription factor, does FOXP1 regulate ATR activation and protect stalled replication fork globally or at any preferred DNA regions?

Response: Thank you for the suggestion. As a transcription factor, FOXP1 has a preference for binding the consensus motif GTAAACA [6, 7]. We conducted in vitro binding assays to test FOXP1's affinity for dsDNA or ssDNA with or without the GTAAACA motif. The results showed that FOXP1 has a stronger preference for

binding dsDNA containing the GTAAACA consensus, while its affinity for ssDNA was similar, regardless of whether the sequence included the GTAAACA motif. This suggests that FOXP1 regulates transcription at specific DNA regions but protects stalled replication forks more broadly. These data are presented in Fig. S2A and Fig. S2B of the revised manuscript.

Figure S2A

Figure S2B

Referee #3:

The authors show that FOXP1 mediates ATR-Chk1 signaling through a mechanism that is controlled by O-GlcNAcylation or phosphorylation of a single serine residue. ATR-Chk1-phosphorylation on FOXP1 serine-396 potentiates the association of RPA-ATR-ATRIP and Chk1 activation. O-GlcNAcylation on FOXP1 serine-396 reduces the association of RPA-ATR-ATRIP. The paper is well written, and the figures are well presented.

Major concern:

Figure 2G shows very little difference between baseline and HU-induced EdU-Biotin FOXP1 PLA foci.

Response: Thank you for pointing this out. We repeated this experiment in Fig. 2G with prolonged HU treatment duration to 2 hours, and included the data in the revised manuscript.

Figure 2G

Can the authors speak to the stoichiometry of FOXP1 phosphorylation vs O-GlcNAcylation vs unmodified serine-396?

Response: Thank you for the suggestion. We examined the O-GlcNAcylation stoichiometry of the FOXP1 phosphorylation mimic mutant S396D and the phosphorylation-defective mutant S396A using the chemoenzymatic labeling method. Immunoprecipitated FLAG-FOXP1 was labeled with GalNAz and conjugated with DBCO-PEG5K. The majority of FOXP1 is modified with O-GlcNAcylation, and the S396D mutant exhibited reduced O-GlcNAcylation compared to the S396A mutant. This data is included in Fig. S4F of the revised manuscript.

Figure S4F

Is there direct evident that endogenous FOXP1 is subject to O-GlcNAcylation in vivo?

I see ectopic expression of tagged proteins in 293T cells.

Response: In Figures 3B and 3F, we demonstrated the O-GlcNAcylation of endogenous FOXP1.

Figure 3B

Figure 3F

Minor concern:

Inconsistent results between Fig 1D and Fig 1H where in the former, FOXP1-2 knockdown has the greatest impact on FOXP1 function, while in the latter, FOXP1-1 has the greatest impact on FOXP1 function. Dose and time titrations are required to speak to activity.

Response: Thank you for your question. We re-performed the experiment in Fig. 1H, extending the HU treatment duration to 2 hours and optimizing the siRNA working concentration.

Figure 1H

The paper would be significantly improved by knockin mutations of FOXP1 alanine-396 or glutamic acid-396.

Response: Thank you for the suggestion. We generated HEK293 cell lines with knock-in mutations of FOXP1 S396A and S396D (as used in the original manuscript) and found that the S396A mutation compromised ATR activation under replication stress. This data replaces the original Fig. S4F and is now presented as Fig. S4G in the revised manuscript.

Figure S4G

Reference

1. Dungrawala, H., et al., *The Replication Checkpoint Prevents Two Types of Fork Collapse without Regulating Replisome Stability*. Mol Cell, 2015. **59**(6): p. 998-1010.
2. Sirbu, B.M., F.B. Couch, and D. Cortez, *Monitoring the spatiotemporal dynamics of proteins at replication forks and in assembled chromatin using isolation of proteins on nascent DNA*. Nat Protoc, 2012. **7**(3): p. 594-605.
3. Bai, G., et al., *HLTF resolves G4s and promotes G4-induced replication fork slowing to maintain genome stability*. Mol Cell, 2024. **84**(16): p. 3044-3060 e11.
4. Bass, T.E., et al., *ETAA1 acts at stalled replication forks to maintain genome integrity*. Nat Cell Biol, 2016. **18**(11): p. 1185-1195.
5. Thompson, J.W., M.E. Griffin, and L.C. Hsieh-Wilson, *Methods for the Detection, Study, and Dynamic Profiling of O-GlcNAc Glycosylation*. Methods Enzymol, 2018. **598**: p. 101-135.
6. Gabut, M., et al., *An alternative splicing switch regulates embryonic stem cell pluripotency and reprogramming*. Cell, 2011. **147**(1): p. 132-46.
7. Li, S., J. Weidenfeld, and E.E. Morrisey, *Transcriptional and DNA binding activity of the Foxp1/2/4 family is modulated by heterotypic and homotypic protein interactions*. Mol Cell Biol, 2004. **24**(2): p. 809-22.

Dr. Xingzhi (Xavier) Xu
School of Medicine, Shenzhen University
Cell Biology
1066 Xueyuan Blvd, Rm. A6-901
Shenzhen University Lihu Campus
Shenzhen, Guangdong 518055
China

5th Nov 2024

Re: EMBOJ-2024-117634R
FOXP1 phosphorylation/O-GlcNAcylation in regulating ATR activation in response to replication stress

Dear Xavier,

Thank you for submitting your revised manuscript to The EMBO Journal. Two of the original referees have now assessed it once again (see comments below), and both of them are overall satisfied with your responses and revisions. We shall therefore be happy to accept the study for publication, pending addressing of the following editorial issues:

- Since we mandate addition of ORCID identifiers for all (co-)corresponding authors, please remind Dr. Peng Gong to obtain and add an ORCID to their author profile in our submission system. This needs to be added by Dr. Gong personally, and we already sent them an email with an URL and instructions.

- Please note that Materials and Methods need to be described in the main text using our 'Structured Methods' format, including a Reagents and Tools Table (listing key reagents, experimental models, software and relevant equipment and including their sources and relevant identifiers) followed by a Methods and Protocols section. A downloadable template (.docx) for the Reagents and Tools Table can be found in our author guidelines:
<https://www.embopress.org/page/journal/14693178/authorguide#structuredmethods>.

- Please correct the nomenclature of the five Expanded View Figures, both in the files, the legends, and the text consistently to "Figure EV1-5". Make sure to include their legends in the main text file, right after the legends for the main figures.

- Please double-check to make sure to all relevant funding information mentioned in the manuscript is also entered into our submission system.

- Please adjust the format of the reference list and of the in-text citations according to EMBO Journal format (alphabetical order, author name et al + year...)

- As we are switching from a free-text author contribution statement towards a more formal statement based on Contributor Role Taxonomy (CRediT) terms, please remove the present Author Contribution section and instead specify each author's contribution(s) directly in the Author Information page of our submission system during upload of the final manuscript. See <https://casrai.org/credit/> for more information.

- Please rename the Conflict of Interest section into "Disclosure and Competing Interests Statement", in accordance with our updated Guide to Authors (<https://www.embopress.org/competing-interests>)

- Please add the header "Data Availability" to the statement that no data has been externally deposited.

- Since we do not allow reference to "data not shown", please either include data or remove the data not shown statement on current page 15.

- Please include at least one reference to Figure 2H, which appears currently missing.

- Please note that source data files need to be saved in a scheme, one figure/folder, and then uploaded as .zip files. E.g. all the Source data files for figure 1 need to be saved in a single folder and this needs to be zipped and then uploaded as "SD figure 1.zip" file. For EV and/or appendix figures, ZIP together all source data.

- Please provide suggestions for a short 'blurb' text prefacing and summing up the study in two sentences (max. 250 characters), followed by 3-5 one-sentence 'bullet points' with brief factual statements of key results of the paper; they will form the basis of an editor-written 'Synopsis' accompanying the online version of the article. Please also upload a synopsis image, which can be used as a "visual title" for the synopsis section of your paper. The image should be in PNG or JPG format with the modest

dimensions of EXACTLY 550 pixels wide and 300-600 pixels high (maybe based on a condensed version of Figure 6?).

- Finally, during routine pre-acceptance checks, our data editors have raised the following queries regarding figures, data, and legends; I would appreciate if you briefly answered to them in the cover letter of your final submission, and made the requested text modifications with changes/additions highlighted via the "Track changes" option, to facilitate our final checking.

* Please note that the exact p values are not provided in the legends of figures 1f-g; 2g; 5g, "supplementary" figures 1d, f; 2g; 5e-f.

* Please note that information related to n is missing in the legend of figure 1g.

* Although 'n' is provided, please describe the nature of entity for 'n' in the legends of figures 1f; 5g, "supplementary" figures 1d, f; 5e-f.

* Please note that the error bars need to be defined in the legends of figures 1g, "supplementary" figures 1d, f; 5e-f.

* Please note that scale bar and its definition are missing for supplementary figure 5a.

* Please note that the asterisk is not defined in the legend of figure 4c, supplementary figure 5c. This needs to be rectified.

I am therefore returning the study to you once more, to allow you to incorporate these final changes and upload all requested files. Once we have received them, we should be able to swiftly proceed with acceptance and production of the manuscript.

With kind regards,

Hartmut

- size of the scale bars that are mandatory for all micrograph panels

- the statistical test used to generate error bars and P-values

- the type error bars (e.g., S.E.M., S.D.)

- the number (n) and nature (biological or technical replicate) of independent experiments underlying each data point

- Figures may not include error bars for experiments with $n < 3$; scatter plots showing individual data points should be used instead.

4) Each main and each Expanded View (EV) figure should be uploaded as individual production-quality files (preferably in .eps, .tif, .jpg formats). For suggestions on figure preparation/layout, please refer to our Figure Preparation Guidelines:

8) Please note that supplementary information at EMBO Press has been superseded by the 'Expanded View' for inclusion of additional figures, tables, movies or datasets; with up to five EV Figures being typeset and directly accessible in the HTML version of the article. For details and guidance, please refer to:

[embopress.org/page/journal/14602075/authorguide#expandedview](https://www.embopress.org/page/journal/14602075/authorguide#expandedview)

9) To facilitate reproducibility and cross-laboratory adoption of methodologies, please structure the Materials & Methods section as outlined in our guide to authors, including a completed Reagents and Tools Table that can be downloaded from our author guidelines as well (<https://www.embopress.org/page/journal/14602075/authorguide#structuredmethods>).

10) Digital image enhancement is acceptable practice, as long as it accurately represents the original data and conforms to community standards. If a figure has been subjected to significant electronic manipulation, this must be clearly noted in the figure legend and/or the 'Materials and Methods' section. The editors reserve the right to request original versions of figures and the original images that were used to assemble the figure. Finally, we generally encourage uploading of numerical as well as gel/blot image source data; for details see: [embopress.org/page/journal/14602075/authorguide#sourcedata](https://www.embopress.org/page/journal/14602075/authorguide#sourcedata)

At EMBO Press, we ask authors to provide source data for the main manuscript figures. Our source data coordinator will contact you to discuss which figure panels we would need source data for and will also provide you with helpful tips on how to upload and organize the files.

In the interest of ensuring the conceptual advance provided by the work, we recommend submitting a revision within 3 months (3rd Feb 2025). Please discuss the revision progress ahead of this time with the editor if you require more time to complete the revisions. Use the link below to submit your revision:

Link Not Available

Referee #1:

The authors addressed all my comments and I believe the manuscript has significantly improved and is now ready for publication.

Referee #2:

The authors have addressed all of my concerns, and I therefore recommend this manuscript for publication

Dr. Xingzhi (Xavier) Xu
School of Medicine, Shenzhen University
Cell Biology
1066 Xueyuan Blvd, Rm. A6-901
Shenzhen University Lihu Campus
Shenzhen, Guangdong 518055
China

18th Nov 2024

Re: EMBOJ-2024-117634R1
FOXP1 phosphorylation/O-GlcNAcylation in regulating ATR activation in response to replication stress

Dear Dr. Xu,

Thank you for submitting your final revised manuscript for our consideration. I am pleased to inform you that we have now accepted it for publication in The EMBO Journal.

With kind regards,

Hartmut
